# The urban green carbon index (UGCI): A spatial framework for suggesting urban carbon management

Inhye Seo[1], Junge Hyun[2,3], Bokyung Son[4], Yeonsu Lee[4], Jungho Im[4], Jongho Kim[5], Sujong Jeong[5], Gayoung Yoo[1,2]*

1 Department of Applied Environmental Science, Kyung Hee University, Yongin, Republic of Korea, 2 Department of Environmental Science and Engineering, Kyung Hee University, Yongin, Republic of Korea, 3 Department of Ecology & Evolutionary Biology, Yale University, Connecticut, United States of America, 4 Department of Civil, Urban, Earth, and Environmental Engineering, Ulsan National Institute of Science and Technology, Ulsan, Republic of Korea, 5 Department of Environmental Planning, Graduate School of Environmental Studies, Seoul National University, Seoul, Republic of Korea

* gayoo@khu.ac.kr

## Abstract

Enhancing carbon (C) sequestration in urban green spaces is essential for urban carbon neutrality; yet the spatial heterogeneity and structural complexity of urban landscapes hinder the development of targeted, actionable management strategies. This study introduces a decision-support framework that integrates a novel Urban Green Carbon Index (UGCI) with a multi-dimensional analytical approach for assessing and guiding urban C management. The UGCI combines 30 m-resolution spatial data on vegetation C storage, soil C storage, net C uptake, and soil C storage potential, and identifies the three most influential components based on the weights derived from their spatial variability. The multi-dimensional approach depicts the relative importance among the three core UGCI components, thereby enabling site-specific and component-focused management directions. A case study in Suwon, South Korea, showed that the UGCI captured pronounced within-patch heterogeneity, recording 21–45% lower values at edges compared with interiors, thereby identifying specific locations within green patches where localized management to improve C conditions is most warranted. Based on the limited components revealed by the ternary analysis, tailored management practices, such as surface litter retention, soil decompaction, and multi-layered planting, were proposed to improve the weaker C components. This framework provides a scientifically grounded and policy-ready basis for spatially explicit diagnosis and management of urban C, supporting a transition from generalized greening efforts toward quantifiable, site-specific C enhancement strategies.

**Data availability statement:** The code and raw dataset generated by the authors and used in this study are publicly available in the first author's GitHub repository: https://github.com/laupran/Calculating-UGCI.git.

**Funding:** This work was supported by the Korea Environmental Industry & Technology Institute (KEITI) through the "Climate Change R&D Project for the New Climate Regime" (Grant No. RS-2022-KE002092, to Sujong Jeong) and the "Project for Developing an Observation-Based GHG Emissions Geospatial Information Map" (Grant No. RS-2023-00232066, to Sujong Jeong), funded by the Ministry of Climate, Energy and Environment (MCEE), Republic of Korea. The funders had no role in study design, data collection and analysis, decision to publish, or preparation of the manuscript.

**Competing interests:** The authors have declared that no competing interests exist.

## 1. Introduction

Although urban areas are responsible for a large share of global carbon (C) emissions, they also hold substantial potential for C sequestration within green spaces and soils, making their management crucial for climate-change mitigation. In 2020, cities accounted for approximately 67–72% of global C emissions, emitting around 29 $GtCO_2eq$ [1]. Long-term projections indicate that under worst-case Representative Concentration Pathway (RCP) scenarios, urban $CO_2$ emissions could comprise up to 77% of the global share, reaching 98.8 $GtCO_2eq$ by 2100 [2]. As the contribution of urban areas to global emissions continues to grow, incorporating urban C sequestration into climate mitigation strategies becomes increasingly important, not only to offset local emissions but also to enhance the overall resilience of urban ecosystems.

Urban green spaces are among the most promising means to achieve this goal [3]. However, most cities face severe land scarcity for new green infrastructure, and existing green spaces are often degraded by anthropogenic disturbances and extreme climate events [4–6]. Moreover, expanding vegetation cover alone may not ensure long-term C sequestration, since the capacity to retain and stabilize C ultimately depends on soil quality, which is often compromised in urban environments [7–9]. In fragmented and heterogeneous urban landscapes, soil physical, chemical, and biological properties vary considerably, resulting in uneven conditions for vegetation growth and soil C stabilization. Such variability constrains the overall effectiveness of C sequestration through factors such as compaction, suboptimal pH, and reduced microbial activity [10–14]. Therefore, strategies that integrate vegetation management with soil-based approaches are essential to maximize the C sequestration potential of urban green spaces.

Given these complexities, a spatially explicit and integrative approach is required to effectively manage urban C storage and sequestration. However, because urban areas exhibit highly diverse characteristics depending on the region, it is extremely difficult to generalize where and what type of interventions for green areas are needed [15]. In practice, resource constraints often limit the extent to which such interventions can be implemented, underscoring the need for spatial prioritization to maximize their effectiveness [16,17]. Therefore, developing a comprehensive decision-support framework is essential for providing scientific evidence to local policymakers, by identifying green areas under poor C condition and guiding site-specific C interventions.

Despite these necessity, most existing studies have conducted broad-scale assessments of C storage, focusing on afforestation site selection or urban land-use zoning at a macro level [18,19]. However, such methods are not well-suited to the highly fragmented nature of urban environments, where C sequestration capacity varies significantly. Furthermore, these approaches often emphasize standing C stocks rather than accounting for the dynamic sequestration processes that shape long-term C retention in the system.

To address these gaps, this study proposes a novel decision-support framework that evaluates current C storage and sequestration while identifying areas with poor C conditions. The framework integrates both vegetation and soil indicators to provide

targeted management strategies. We apply this framework to Suwon, a rapidly urbanizing metropolitan city in South Korea with diverse land cover types. Our analysis assesses vegetation and soil C storage, net C uptake, and soil C sequestration potential across the city, culminating in the calculation of the urban green carbon index (UGCI). By integrating these four key components, our study offers a practical yet scientifically robust tool for policymakers, enabling targeted management strategies to significantly enhance C sequestration across urban landscapes.

## 2. Materials and methods

### 2.1. Decision-support framework for urban C management

To efficiently identify and remediate urban green spaces with poor C storage and sequestration, this study proposes a two-step decision-support framework (Fig 1): (1) diagnosis step and (2) prescription step.

In the diagnosis step, we calculated the urban green carbon index (UGCI) on a grid basis across the urban landscape to identify the grids with severely poor C storage and sequestration conditions. In the prescription step, we further decomposed the UGCI into each component and visualized them using multi-dimensional plot after determining the grids requiring management. This step identifies whether vegetation- or soil-related constraints dominate C storage and sequestration

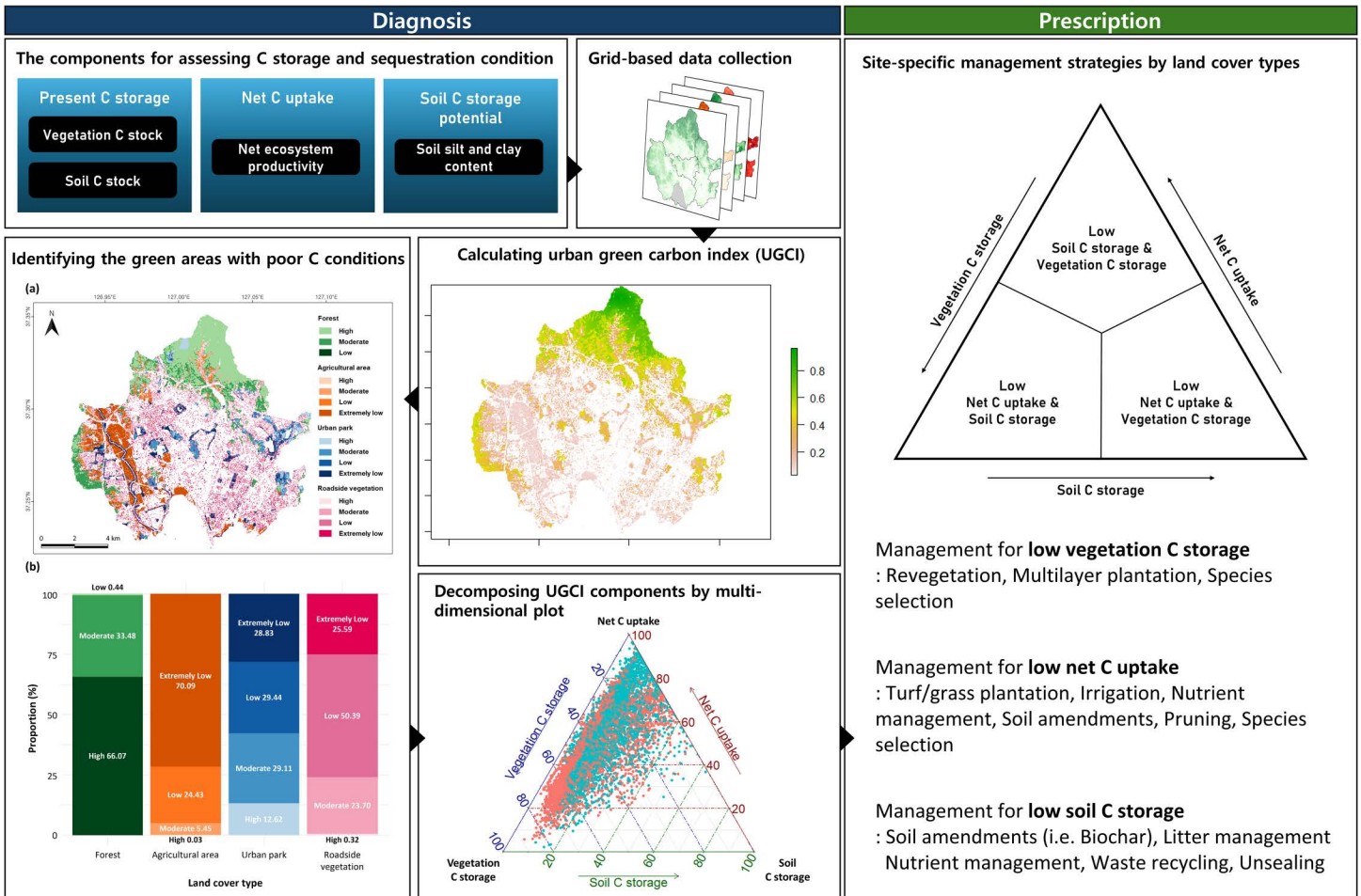

**Fig 1. A two-step decision-support framework using the urban green carbon index (UGCI) and the multi-dimensional plot approach.**

in each grid. Based on the results, tailored management strategies, including vegetation restoration, soil amendments, and land-use modification, were formulated to address the limiting factors. The proposed strategies do not incorporate cost-benefit or opportunity-cost analyses; these limitations and the corresponding directions for future research are discussed separately in the discussion section.

**2.1.1. Diagnosis of urban green areas: Urban green carbon index (UGCI).** The UGCI quantifies overall C storage and sequestration conditions at the grid level (e.g., 30 × 30 m). It integrates four key components, each representing distinct but complementary aspects of urban C dynamics:

- Vegetation C storage: The total above- and belowground vegetation C stocks within each grid,

- Soil C storage: The total amount of organic C stored in the soil within each grid,

- Net C uptake: The net ecosystem productivity (NEP) of each grid, calculated as the difference between gross primary productivity (GPP) and ecosystem respiration ($R_{eco}$),

- Soil C storage potential: An indicator of the soil's inherent capacity to accumulate and stabilize C, primarily represented by the soil's silt and clay content per grid [20,21].

To ensure comparability across components with different units, a min-max normalization method was applied, scaling all values to a range between 0 and 1. The normalized value of the component $j$ in grid $i$ ($X'_{ij}$) is given by:

$$X'_{ij} = \frac{X_{ij} - X_{j\,min}}{X_{j\,max} - X_{j\,min}}$$

(1)

where $X_{ij}$ indicates the value of component $j$ in grid $i$, $X_{j\,min}$ and $X_{j\,max}$ indicate the minimum and maximum values of component $j$ among all grids ($i \ni [1, n]$; $j \ni [1, m]$). Here, n denotes the total number of grids within the study area, and m is the number of components.

The UGCI for each grid $i$ ($UGCI_i$) was computed by using a linear weighted analysis, following the equation [22]:

$$UGCI_i = \sum_{j=1}^{m} X'_{ij} W_j$$

(2)

where $X'_{ij}$ is the min-max normalized value of component $j$ in grid $i$, and $W_j$ indicates the weights for the component $j$ ($i \ni [1, n]$; $j \ni [1, m]$). n is the total number of grids, and m is the total number of components.

To objectively determine the weights of each UGCI component, an entropy-based weighting method, which is commonly used in the multi-criteria decision-support process, was adopted [23]. Here, entropy indicates a spatial heterogeneity of each component. A higher entropy value indicates greater spatial variability, providing more significant information for decision-support, thus resulting in a higher weight for the component. Conversely, components with low spatial variability (lower entropy) receive lower weights. The entropy-based weights ($W_j$) were calculated using the following formulas [22,24]:

$$W_j = \frac{1 - E_j}{\sum_{j=1}^{m} (1 - E_j)}$$

(3)

where the information entropy ($E_j$) of component $j$ is defined as:

$$E_j = -k \sum_{i=1}^{n} P_{ij} ln(P_{ij}) = -\frac{1}{lnn} \sum_{i=1}^{n} P_{ij} ln(P_{ij})$$

(4)

with a constant $k$, usually equal to $1/ln(n)$ so that larger than 0, and $P_{ij}$ representing the proportion of component $j$ in grid $i$ ($i \ni [1, n]; j \ni [1, m]$):

$$P_{ij} = \frac{X'_{ij}}{\sum_{i=1}^{n} X'_{ij}}$$

(5)

where $X'_{ij}$ is the min-max normalized value of component $j$ in grid $i$ ($i \ni [1, n]; j \ni [1, m]$).

According to Eq 2, a lower UGCI value indicates severely poorer conditions regarding vegetation C storage, soil C storage, net C uptake, and soil C storage potential. After computing UGCI values for all grids, we categorized them into four distinct management levels using a quartile-based classification: 'extremely low', 'low', 'moderate', and 'high'. Grids classified as 'extremely low' or 'low' represent areas requiring immediate and targeted interventions to effectively improve their C storage and sequestration.

**2.1.2. Prescribing site-specific intervention.** For grids with 'extremely low' and 'low' UGCI, we further decomposed the UGCI into individual components to recommend the targeted management. To effectively visualize the main factors contributing to poor C storage and sequestration, we selected the components with the three highest weighting values to construct a ternary plot. Each grid's position within a ternary plot was determined based on the relative proportions of these three selected components, calculated as follows:

$$\begin{cases} X = \frac{a}{a+b+c} \\ Y = \frac{b}{a+b+c} \\ Z = \frac{c}{a+b+c} \end{cases}$$

(6)

where $X$, $Y$, and $Z$ represent the relative proportions of the three components which will be shown in the ternary plots, while $a$, $b$, and $c$ represent the min-max normalized values of each component. ($0 \le a, b, c \le 1$)

Points located near a vertex of the ternary plot indicate grids where the two opposite components are substantially weaker, thereby suggesting clear management directions. In contrast, points clustered near the center represent grids with uniformly low values across all three components. These spatial patterns allow intuitive identification of the component(s) that require greater attention to improve C storage and sequestration conditions. Accordingly, site-specific management strategies were derived from the relative position of each grid within the ternary plot.

## 2.2. Case study

Through the case study, we demonstrate the applicability of the proposed framework and evaluate the practical availability of UGCI in an actual urban environment. Our study site was Suwon (37°17'28"N, 127°00'32" E), a representative metropolitan city in South Korea with a population density of 9,884 persons km$^{-2}$ as of 2024 (Korean Ministry of the Interior and Safety). The city has a temperate monsoon climate with a mean annual temperature of 12.5 °C and precipitation of 1,320.3 mm (Korean Meteorological Administration). Although past urbanization has reduced much of its natural green cover, Suwon has actively expanded and restored urban green spaces since the early 2000s. Today, diverse green patches—including urban forests, parks, and roadside vegetation—are interspersed among dense urban structures, providing an ideal setting to evaluate the framework's applicability. In this case study, the data collection was conducted for the year 2020. All spatial data layers were resampled to a 30 m resolution, balancing the finest available resolution of input datasets with the spatial granularity required for urban-scale analysis.

**2.2.1. Land cover classification.** We used a high-resolution (1 m) subdivision land-cover map provided by the Environmental Geographic Information Service of the Korean Ministry of Environment. The map initially classified the area into ten categories: three forest types (broadleaf, coniferous, and mixed forest), agricultural area, grassland, wetland, bare

land, impervious area, water, and others. The dataset, originally in vector format, was converted to a raster at 1 m spatial resolution for further processing.

Because this base map did not separately delineate urban parks and roadside vegetation, we refined it using supplementary datasets. Urban parks were identified from the park boundary data of the Korean Ministry of Land, Infrastructure and Transport [25], and the roadside vegetation was mapped using a deep-learning classification model based on the U-Net algorithm [26]. The U-Net model employed airborne LiDAR and high-resolution (0.25 m) RGB orthogonal imagery to detect tree-covered areas along roads. The model successfully identified an additional 13.69% of tree-covered areas that had not been classified in previous datasets, achieving a classification accuracy of 96.29% (Text S1 in S1 File). This refinement was essential for accurately representing highly fragmented urban vegetation often overlooked at coarser resolutions (>1 m).

Finally, all refined datasets were combined to produce an updated land-cover map at 1 m resolution, comprising ten classes: broadleaf forest, coniferous forest, mixed forest, agricultural area, urban park, roadside vegetation, impervious area, bare land, water, and others (Fig 2). Using the updated 1 m land-cover map, we aggregated it to a 30 m grid by assigning each grid cell the dominant land-cover type when multiple classes were present. Grasslands and wetlands, which together occupy less than 1% of the total area and are mainly used as riverside parks or golf courses, were included in the urban-park category for analysis.

**2.2.2. Vegetation and soil C storage.** We estimated vegetation and soil C storage for each 30 m grid. For vegetation C storage, we used Eq 7, following the 2006 IPCC Guidelines, to compute vegetation C storage at a 1 m resolution and then aggregated the results to a 30 m grid by summation [27]:

$$VCS = A \times V \times BEF \times D \times (1+R) \times CF \qquad (7)$$

where $VCS$ is the vegetation C storage (tonnes C; tC), $A$ is the vegetated area (ha) within the grid, $V$ is the growing stock volume (m$^3$ ha$^{-1}$), $BEF$ is the biomass expansion factor, $D$ is the basic wood density (tonnes biomass m$^{-3}$), $R$ is the ratio of belowground biomass to aboveground biomass, and $CF$ is the C fraction in biomass (tC (tonnes biomass)$^{-1}$).

The vegetated area ($A$) was derived for each land cover type based on the updated land-cover map. The growing stock volume ($V$) for Suwon was obtained from the 2020 forest basic statistics [28]. The values for $BEF$, $D$, $R$, and $CF$ were the country-specific coefficients [29].

For the roadside vegetation where a single tree is separately planted, a different method was applied to calculate vegetation C storage. First, we applied Eq 7 for the roadside vegetation within each 30 m grid using the coefficients (i.e., $V$, $BEF$, $D$, and $R$) corresponding to natural forests and multiplied the correction factor 0.8 reflecting the difference between natural and urban trees [30]. To enhance the accuracy of this estimation, we developed a machine-learning model to correct the bias between the values from Eq 7 and the ground-truth roadside vegetation C storage. The ground-truth values were calculated from directly measured tree counts and diameter at breast height (DBH) for 69 representative roadside vegetation grids in Suwon using Eq 8:

$$VCS_t = \sum_{j=1}^{M} \sum_{i=1}^{N_j} a_j \times (DBH_{ij})^{b_j} \qquad (8)$$

where $VCS_t$ is the ground-truth roadside vegetation C storage within each 30 m grid (tonnes C; tC), $DBH_{ij}$ is the diameter at breast height (cm) for individual tree $i$ of the tree species $j$, $N_j$ is the number of trees of species $j$, and $M$ is the number of tree species within the grid. $a_j$ and $b_j$ are the species-specific allometric coefficients [31].

The roadside vegetation C storage from Eq 7 was adjusted using the difference between the values from Eq 7 and Eq 8, which were derived from the machine-learning model using the tree height and intensity from airborne LiDAR,

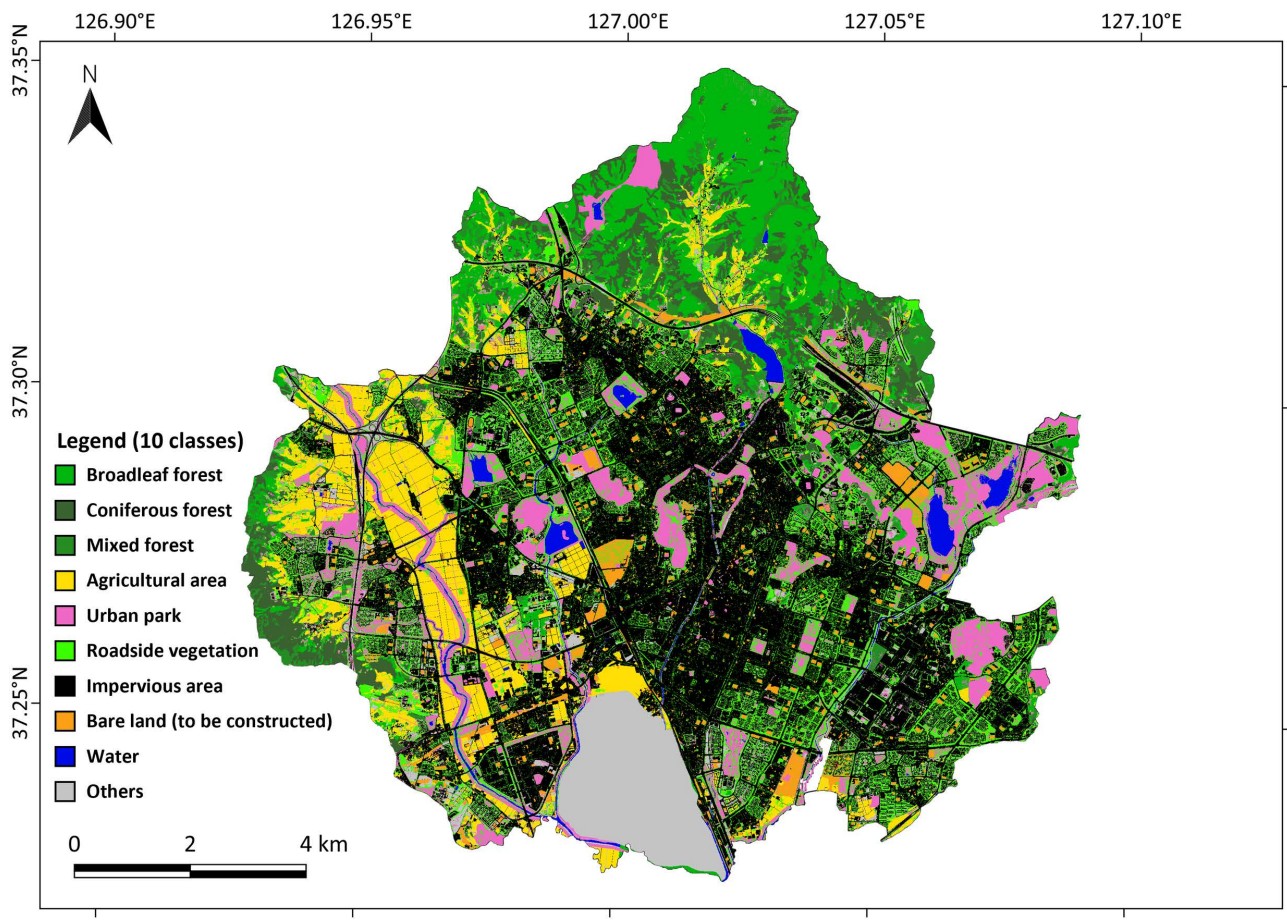

**Fig 2. The refined high-resolution land-cover map of the study area (Republished from V-world data platform under a CC-BY license, with permission from V-world data platform (2025)).**

normalized difference vegetation index (NDVI) from Sentinel-2, and the Eq 7 values as input variables. The 10-fold cross-validation using 69 ground truth values achieved a coefficient of determination ($R^2$) of 0.81 and a root mean square error (RMSE) of 5.39 tC ha$^{-1}$. The detailed information for estimating roadside vegetation C storage can be found in Lee et al. (2024) and Text S2 in S1 File. The final vegetation C storage map was produced by integrating the vegetation C storage values of all land-cover types within each 30 m grid.

Soil C storage data were obtained from the Global Soil Organic Carbon (GSOC) map (version 1.6.1, 1 km spatial resolution) [32,33]. To ensure data reliability and suitability for our analysis, we validated the GSOC map using soil C storage measurements derived from 121 legacy soil samples collected from diverse land-cover types across South Korea during 2020–2024 by the Kyung Hee University Carbon Neutrality Technology Center [34,35]. The validation showed a high coefficient of determination ($R^2 = 0.84$), confirming that the reliability of the product for our purpose. Additional details on data accuracy and uncertainty are available in the GSOC map v1.6 Technical Report [32]. To unify the spatial resolution with other datasets, we resampled the map to a 30 m resolution using a bilinear interpolation method, preserving its continuous gradient.

**2.2.3. Net C uptake: net ecosystem productivity (NEP).** Net C uptake, which is represented by net ecosystem productivity (NEP), was calculated for each 250 m grid as the difference between gross primary productivity (GPP)

and ecosystem respiration ($R_{eco}$). Then, the spatial resolution was downscaled to a 30 m grid basis using a bilinear interpolation method to preserve the continuous data gradient.

GPP was estimated using a random forest model based on the light-use efficiency (LUE) concept on a daily basis [36]. The model structure and choice of predictors followed the vegetation photosynthesis and respiration model, which provides a simple yet robust parameterization of gross ecosystem exchange and net ecosystem exchange through the use of enhanced vegetation index (EVI), land surface water index (LSWI), and meteorological scalars [36]. The key input variables for this model included air temperature and vapor pressure deficit (VPD) from the Local Data Assimilation and Prediction System (LDAPS) provided by Korea Meteorological Administration, photosynthetically active radiation (PAR) from HIMAWARI-8, and EVI and LSWI from MODIS product. Further details on the datasets used are provided in Table 1. The GPP ($\mu$mol $CO_2$ m$^{-2}$ s$^{-1}$) estimation formula is as follows:

$$GPP = LUE \times nPAR \times fPAR \times (0.382 \times T_{air} + 0.0905 \times VPD + 17.72) \times \left( \frac{1 + LSWI}{1 + LSWI_{max}} \right) \tag{9}$$

where $LUE$ is light use efficiency ($\mu$mol $CO_2$ $\mu$mol$^{-1}$ PPFD; Photosynthetic Photon Flux Density), $nPAR$ is normalized PAR (dimensionless), $fPAR$ is fraction of absorbed PAR by vegetation canopy derived from EVI (dimensionless), $T_{air}$ is air temperature (°C), $VPD$ is vapor pressure deficit (kPa), $LSWI$ is land surface water index calculated by using NIR and MIR (dimensionless), and $LSWI_{max}$ is the maximum LSWI within the plant-growing season (dimensionless) for each grid.

The GPP model was trained using the FLUXNET2015 dataset [37] and validated against local flux tower measurements at Mt. Taehwa in South Korea (37°18′15″ N, 127°18′50″ E), achieving an $R^2$ of 0.76 and an RMSE of 0.41 $\mu$mol $CO_2$ m$^{-2}$s$^{-1}$. Although Mt. Taehwa is located outside our study site, the temporal variation patterns of its GPP and $R_{eco}$ data were consistent with those observed at the eddy covariance towers in Mt. Nam and Changgyeong Palace, representative urban green areas in Seoul.

Hourly ecosystem respiration ($R_{eco}$) was modeled as a function of the GPP accumulated during the 7 days preceding a specific date ($GPP_{acc}$) and the air temperature ($T_{air}$), reflecting the impact of vegetation productivity on respiration rates (Eq 10) [38,39]. The model parameters were initialized within the empirical ranges suggested by the existing studies [36,39] and subsequently optimized using the FLUXNET2015 data and meteorological data to minimize the residuals between modelled and observed $R_{eco}$.

$$R_{eco} = (1 + 0.20 \times GPP_{acc}) \times \exp(1.16 \times \frac{T_{air}}{\max(T_{air})}) \tag{10}$$

where $R_{eco}$ is the ecosystem respiration ($\mu$mol $CO_2$ m$^{-2}$ s$^{-1}$), $GPP_{acc}$ is the accumulated GPP ($\mu$mol $CO_2$ m$^{-2}$ s$^{-1}$), and $T_{air}$ is the air temperature (°C).

Similar to the GPP model, the $R_{eco}$ model was fitted on the FLUXNET2015 dataset and validated with the same flux tower measurements from Mt. Taehwa. The validation resulted in an $R^2$ of 0.83 and an RMSE of 0.18 $\mu$mol $CO_2$ m$^{-2}$ s$^{-1}$, confirming that the NEP values are appropriate and reliable for use within our study area. The calculated NEP values were integrated to annual totals and converted into units of tonnes of C per hectare (tC ha$^{-1}$ yr$^{-1}$).

**2.2.4. Soil C storage potential: Silt and clay content.** Given that soil silt and clay content play a critical role in stabilizing C [40,41], we used silt and clay content as a feasible proxy indicator of physical C storage potential. If a more direct indicator for C storage potential, such as soil C saturation deficit or stabilization capacity, becomes available at a spatially relevant scale, the soil C storage potential of UGCI could be updated to incorporate those direct metrics. While silt and clay content is intrinsic and relatively stable in natural soils, urban soils—particularly those in parks and roadside vegetation—are often reconstructed and sand-dominated, allowing urban soil texture to be modified during construction. Other edaphic factors, such as pH and microbial activity, were not incorporated because their direct relationships with soil C storage potential remain uncertain.

**Table 1. Detailed information of the input data used in model for estimating net C uptake and soil silt and clay content.**

| Model | Category | Input variable | Description | Spatial resolution | Temporal resolution | Source |
|---|---|---|---|---|---|---|
| LUE model for estimating gross primary productivity and ecosystem respiration | Climate | $T_{air}$ | Air temperature from 2 m above the surface | 1.5 km | 3 hrs | LDAPS (https://data.kma.go.kr/data/rmt/rmtList.do?code=340&pgmNo=65) |
| | | VPD | Vapor pressure deficit | 1.5 km | 3 hrs | |
| | | PAR | Photosynthetically active radiation | 1 km | hourly | HIMAWARI-8 |
| | Vegetation | EVI | Enhanced vegetation index | 250 m | 8 days | MOD13Q1.v006 |
| | | LSWI | Land surface water index | 250 m | 8 days | |
| LGBM model for estimating soil silt and clay content | Climate | MAT_3 | Mean annual temperature (past 3 years) | 500 m | hourly | Convergence meteorological dataset provided by Korean Meteorological Administration |
| | | MAT_20 | Mean annual temperature (past 20 years) | 500 m | hourly | |
| | | MAP_3 | Mean annual precipitation (past 3 years) | 500 m | hourly | |
| | | MAP_20 | Mean annual precipitation (past 20 years) | 500 m | hourly | |
| | | Snow_3 | Mean snowfall (past 3 years) | 500 m | daily | |
| | | Solar_rad_3 | Mean solar radiation (past 3 years) | 2 km | hourly | |
| | Vegetation | GPP | Mean gross primary production for recent 3 years | 30 m | hourly | This study |
| | Land cover | IA | Impervious area ratio | 30 m | – | This study |
| | | Road_loc | Location of roadside vegetation (binary) | 30 m | – | |
| | Terrain | Elevation | Elevation | 30 m | – | 30 m DEM data from Korean national geographic information institute |
| | | | Slope | Slope | 30 m | – |
| | | | Aspect | Aspect | 30 m | – |
| | | TPI | Topographic position index | 30 m | – | |
| | | TWI | Topographic wetness index | 30 m | – | |

To generate a 30 m-resolution map of soil silt and clay content, we applied digital soil mapping techniques [42], using 225 legacy soil samples collected by the Carbon Neutral Technology Center, Kyung Hee University. The silt and clay fractions of these samples were determined using the standard hydrometer method [43]. Spatial predictors included climate data from the Korean Meteorological Administration, impervious area ratio and location of roadside vegetation derived from the updated land-cover map, and terrain variables obtained from 30 m digital elevation data provided by the Korean National Geographic Information Institute. Three machine-learning algorithms—random forest, artificial neural network, and light gradient boosting—were evaluated using these predictors (Table 1). The performance of each model was assessed using a 10-fold cross-validation approach. The optimized hyperparameter by the Bayesian optimization method and the performance of optimized models were described in Table S1 in S1 File. The light gradient boosting model performed best, yielding an $R^2$ of 0.84 and an RMSE of 10.33%.

**2.2.5. Statistical and spatial data analysis.** The statistical metrics for vegetation C storage, soil C storage, net C uptake, soil C storage potential, and UGCI by land cover were computed using the '*zonal*' function from the '*raster*'

package in R software (version 4.3.1). Results were visualized using QGIS software (v3.34.13-Prizren). Statistical significance of differences in UGCI and its components among land cover types was assessed using t-tests and ANOVA. To investigate the relationships between the vegetation and soil C components, we performed Pearson's correlation analysis using the '*corr*' package in R software (version 4.3.1).

To evaluate whether UGCI accurately captures variations in C storage and sequestration conditions within complex urban landscapes, we conducted an edge-effect analysis. The "edge effect" refers to altered ecological characteristics occurring at boundaries between different land-cover types [44]. Edges were defined as all grids located within 30 m from patch boundaries, identified using the '*clump*' and '*boundaries*' functions in the '*raster*' package.

For urban parks, which exhibit pronounced fragmentation, we analyzed the relationship between patch size and UGCI using Pearson's correlation, performed with the '*corr*' package in R software (version 4.3.1). Ternary plots illustrating which UGCI components require targeted improvements were generated using the '*ggtern*' package in R software (version 4.3.1).

## 3. Result and discussion

### 3.1. Spatial distribution of vegetation C storage, soil C storage, net C uptake, and soil C storage potential in urban green areas

Vegetation C storage at the grid scale ($30 \times 30$ m) ranged from 0 to 109.22 tC ha$^{-1}$ (Fig 3a). The highest vegetation C storage was observed in the broadleaf forests of Mt. Gwanggyo (109.22 tC ha$^{-1}$), located in the northern part of Suwon. This value is slightly higher than the typical range (~76 tC ha$^{-1}$) reported for urban forests in previous studies [45,46]. Conversely, low vegetation C storage was predominantly observed in the fragmented urban green spaces, particularly roadside vegetation. The average vegetation C storage in the roadside vegetation (9.44 tC ha$^{-1}$) was only 10.90% of the forest average (86.59 tC ha$^{-1}$), primarily due to lower planting densities.

Soil C storage ranged between 24.35 and 52.57 tC ha$^{-1}$ (Fig 3b), which is slightly lower compared to values reported for urban soils in other studies (28–103 tC ha$^{-1}$) [47,48]. The spatial distribution of soil C storage closely mirrored that of vegetation C storage, with maximum values in the northern forest areas and minimum values observed in the roadside vegetation, particularly near the urban core. Because soil C storage results from a balance between vegetation-derived C inputs and microbial decomposition, it demonstrated a strong positive correlation with both vegetation C storage and net C uptake (Pearson's correlation coefficients = 0.63 and 0.69, respectively; $p < 0.001$).

Net C uptake, represented by NEP, ranged from −0.69 to 16.95 tC ha$^{-1}$ yr$^{-1}$ (Fig 3c). The highest value was also observed in the broadleaf forests of Mt. Gwanggyo (16.95 tC ha$^{-1}$ yr$^{-1}$), consistent with previous reports on urban forests [49,50]. Roadside vegetation displayed the lowest values, including net C emissions (negative NEP values), particularly near the city center. This result suggests environmental stressors in central urban areas, such as heat wave during summer or extremely dry soil during winter, are deteriorating roadside vegetation C storage and sequestration [51–53]. The average NEP in urban parks (6.72 tC ha$^{-1}$ yr$^{-1}$) was slightly higher than that in roadside vegetation (5.56 tC ha$^{-1}$ yr$^{-1}$) but substantially lower (54.81%) compared to forests (12.26 tC ha$^{-1}$ yr$^{-1}$). Considering that the tree ages in the urban parks are younger than in the natural forests, a low net C uptake in the urban parks can be partially attributed to a poor photosynthesis due to the less favorable growing conditions.

Soil silt and clay content, an indicator of soil C storage potential, showed variability primarily driven by anthropogenic influences rather than natural soil heterogeneity (Fig 3d). The average soil silt and clay content by land cover types did not show significant differences, representing 33.15% (± 3.58) for the forests, 33.25% (± 5.79) for the urban parks, 36.42% (± 4.93) for the agricultural areas, and 31.85% (± 7.15) for the roadside vegetation. However, the markedly low silt and clay content (~19%) observed in some roadside vegetation soils indicates that these soils are more representative of artificial soils intentionally mixed with coarse sand during the construction of roadside vegetation.

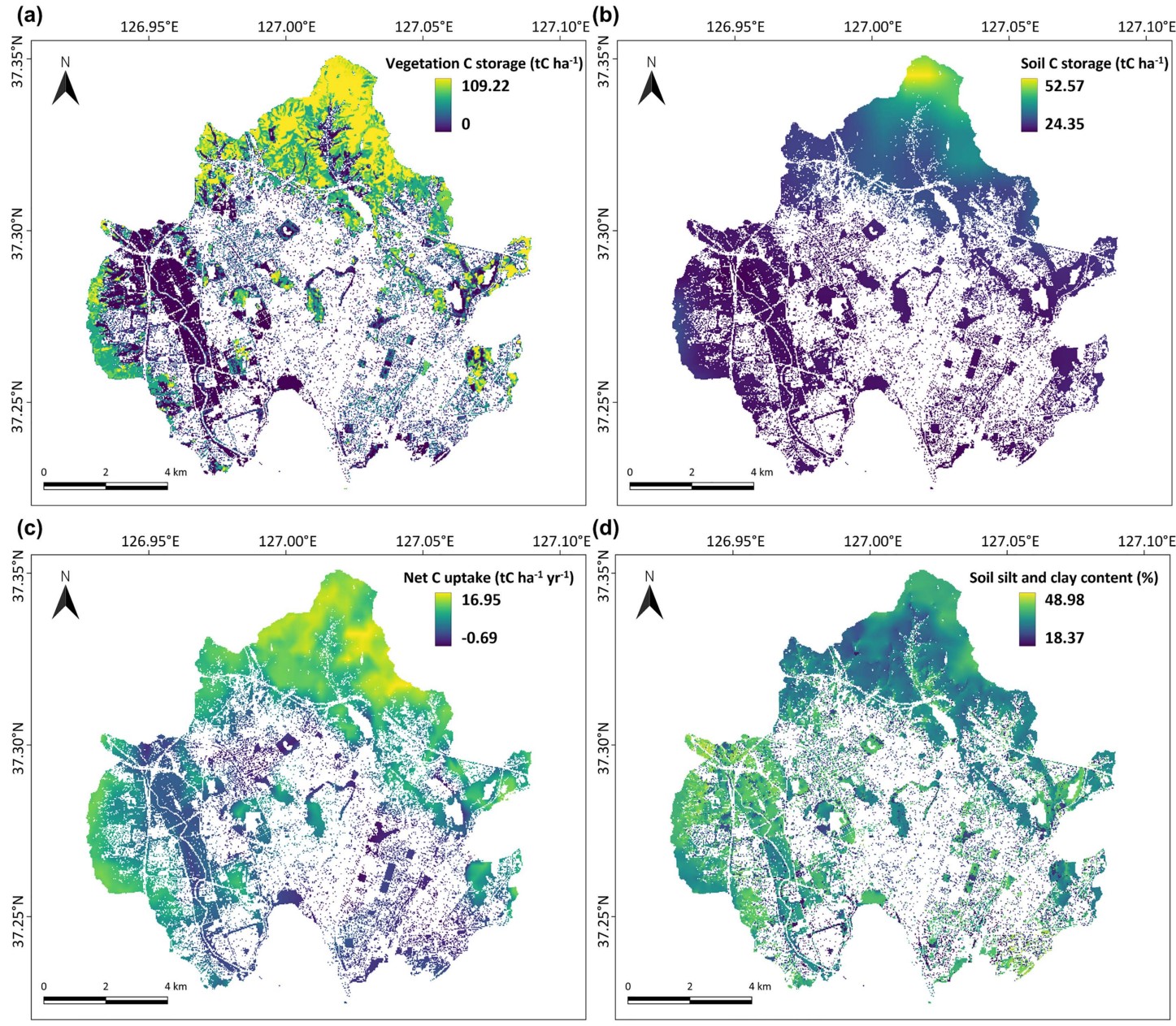

**Fig 3. 30 m resolution maps of (a) vegetation C storage, (b) soil C storage, (c) net C uptake, and (d) soil C storage potential in Suwon's urban green areas.**

## 3.2. Urban green carbon index (UGCI)

Fig 4a illustrates the spatial distribution of UGCI, categorized into four levels by quartile: 'extremely low' (<0.14), 'low' (0.14–0.26), 'moderate' (0.26–0.49), and 'high' (≥0.49). The average UGCI was highest in forests (0.57), followed by urban parks (0.25), roadside vegetation (0.21), and agricultural areas (0.14). The notably low UGCI observed in agricultural areas likely resulted from intensive farming practices, which constrain C accumulation through frequent harvesting

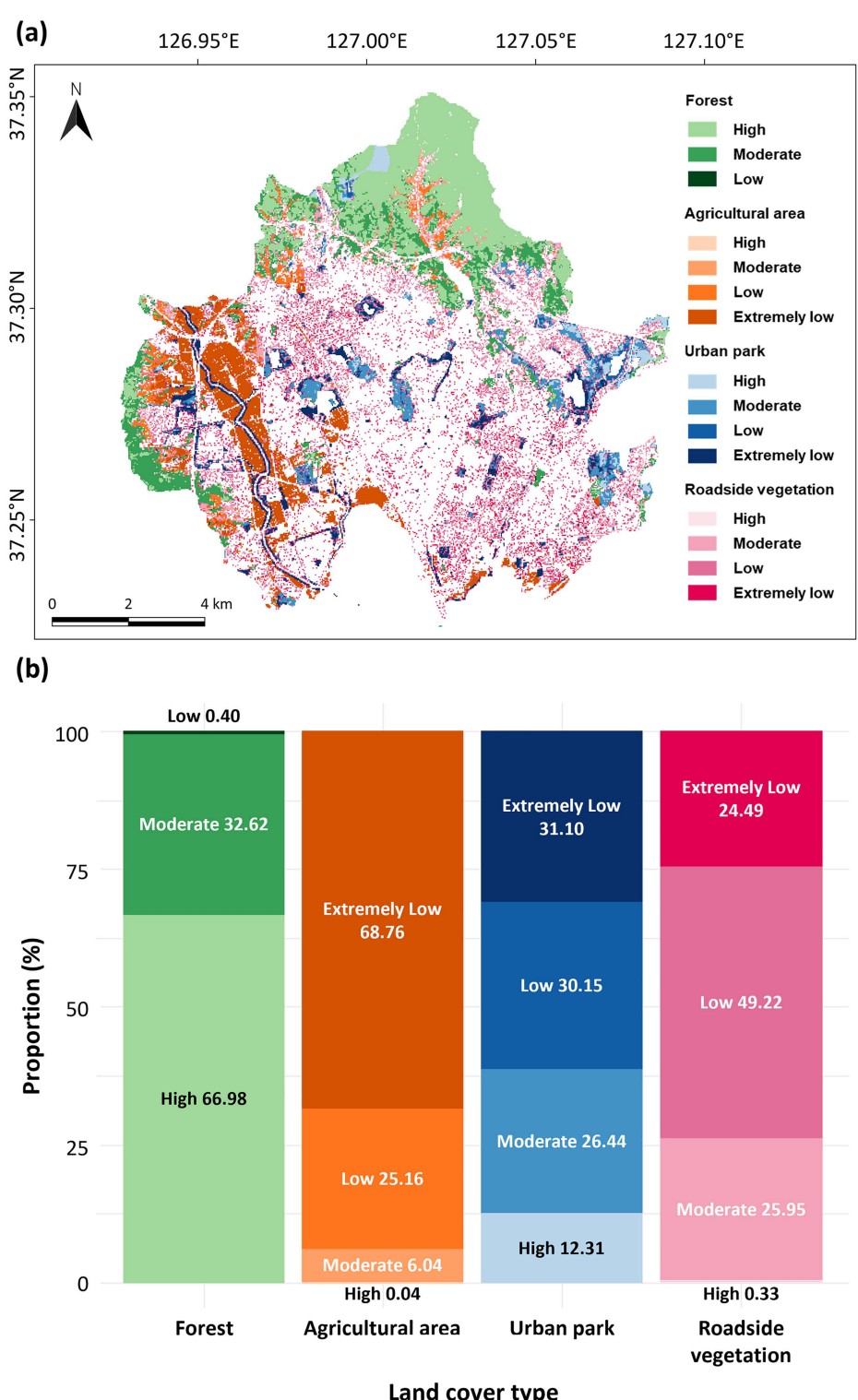

**Fig 4. (a) Spatial distribution and (b) the proportion of UGCI levels by land cover types.**

and soil disturbance. While the average UGCI values for urban parks and roadside vegetation were similar, their proportions across UGCI levels differed significantly (Fig 4b). Roadside vegetation exhibited a 1.63 times higher proportion in the 'low' UGCI level than urban parks, despite marginal differences in the 'extremely low' category. This indicates roadside vegetation requires more widespread attention to enhance C management effectively.

UGCI was particularly effective in identifying poor conditions within land cover types. Edges of the forests and urban parks consistently exhibited 21% and 45% lower UGCI values, respectively, compared to non-edge areas ($p < 0.001$; Fig 5a). Low UGCI values at edges suggest reduced C storage and sequestration capacity, which previous studies attribute to elevated air and soil temperatures, lower humidity, and reduced soil moisture, as well as anthropogenic disturbances like litter removal, air pollution, and human traffic [54–61]. Moreover, the forest and urban park edges adjacent to impervious surfaces (e.g., roads and buildings) demonstrated 18% and 30% lower UGCI compared to those adjacent to non-impervious green areas (Fig 5b). This aligns with prior findings, suggesting impervious surfaces amplify edge effects by exacerbating environmental stress and disturbance, thereby further degrading C storage and sequestration [62–64].

Our analysis also captured how urban structure and configuration influence roadside vegetation's UGCI. Older downtown areas, such as Paldal-gu ($0.29 \pm 0.11$), displayed significantly lower UGCI values compared to newer urban districts like Jangan-gu ($0.35 \pm 0.12$) and Gwonseon-gu ($0.32 \pm 0.13$) ($p < 0.001$; Fig S1 in S1 File). Restrictions due to historical preservation in Paldal-gu resulted in isolated, older roadside vegetation. In contrast, Jangan-gu and Gwonseon-gu's recent urban developments included densely planted, multi-layered vegetation arrangements, effectively increasing UGCI. This finding aligns with previous studies emphasizing that structured, connected green spaces enhance urban C storage and sequestration [34,65].

Additionally, the correlation between the areas of green patches in urban parks (logarithmic scale) and their mean UGCI was very weak ($r = 0.30$, $p < 0.001$; Fig S2 in S1 File), in contrast to previous studies that reported significant positive relationships between park size and vegetation or soil C stocks [66,67]. This weak relationship may reflect the high heterogeneity among urban parks. Some urban parks are intentionally designed as sports fields or open grasslands, where vegetation density is deliberately kept low for recreational or aesthetic purposes. This heterogeneity is also visually

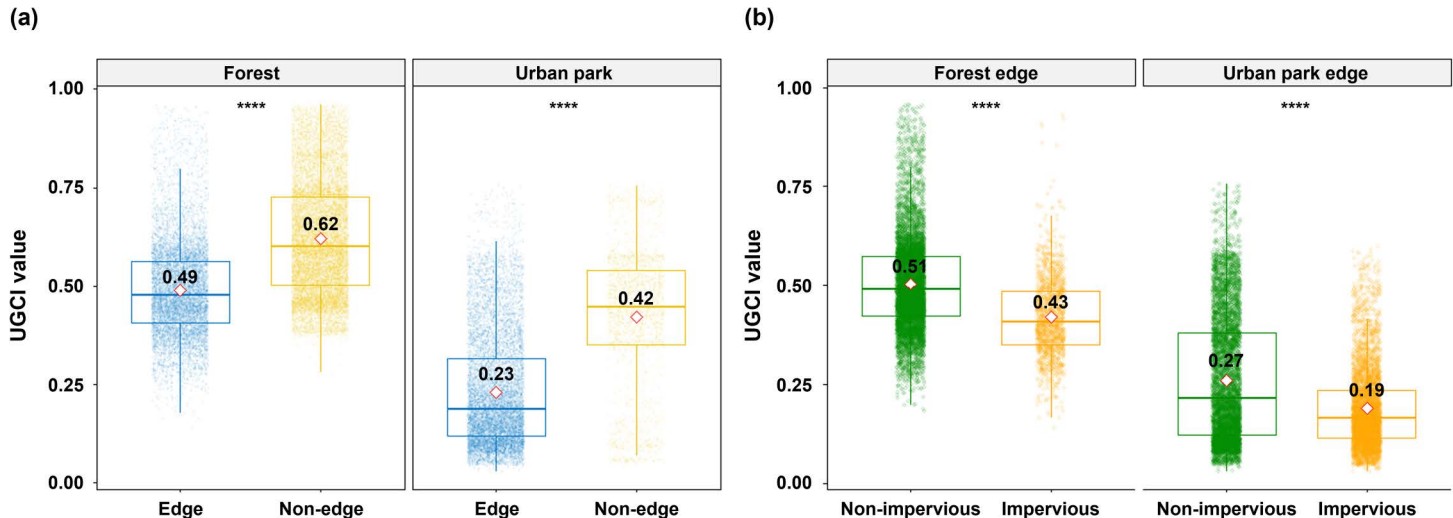

**Fig 5. (a) UGCI boxplot for edge versus non-edge in the forests and the urban parks; (b) UGCI boxplot for the forest edges and the urban park edges according to their adjacent land cover.** Red diamonds indicate the mean UGCI for each group. Statistical significance is denoted by asterisk notation (****, $p < 0.001$).

evident in the spatial distribution and proportional patterns of UGCI shown in <u>Figs 3a</u> and <u>3b</u>, where substantial variation is observed among parks of similar size.

### 3.3.  Site-specific C management strategies and policy implications

<u>Fig 6</u> presents ternary plots illustrating grids classified as 'low' and 'extremely low' UGCI levels, differentiated by land cover type. Among the components of the UGCI, the soil silt and clay content had the lowest weighting factor (0.08), indicating the smallest contribution to the overall variation of the index; therefore, it was excluded from the ternary plot analysis. While our analysis here concentrates on the 'low' and 'extremely low' UGCI levels, it is important to note that a high UGCI does not imply the absence of management needs. Conservation-oriented strategies aimed at protecting existing C stocks and avoiding disturbances may be more appropriate than aggressive intervention in such areas.

In forests, all grids classified as 'low' UGCI were spatially situated at edge areas, emphasizing the need for edge-focused management. These edge grids consistently exhibited the lowest scores for soil C storage, indicating that management efforts should primarily focus on enhancing soil C stability. Because forest-edge soils are more exposed to higher temperatures and direct sunlight, leading to accelerated decomposition [54,68], establishing shade-tolerant native understory shrubs/groundcovers and avoiding intensive leaf litter removal in edge areas can help minimize soil exposure and sustain soil moisture.

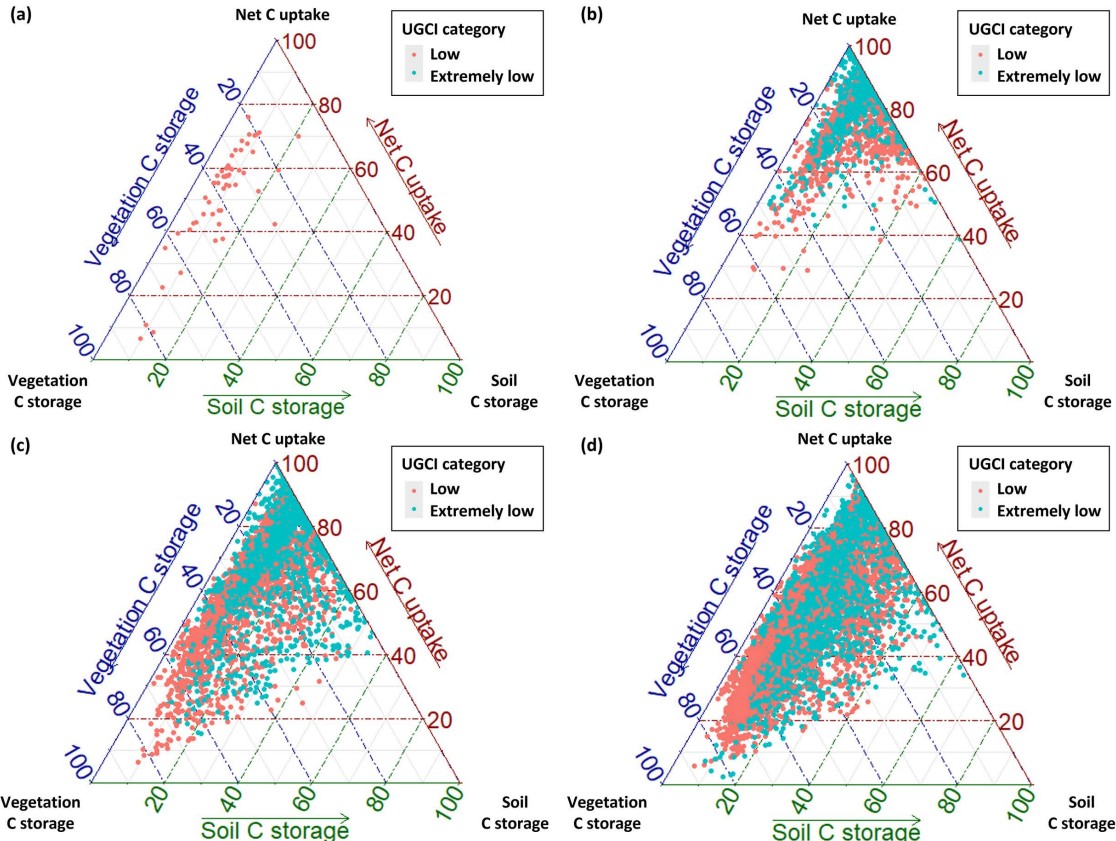

**Fig 6.  Ternary plots depicting 'low' (red points) and 'extremely low' (blue points) UGCI levels across land cover types: (a) forests, (b) agricultural areas, (c) urban parks, and (d) roadside vegetation.**

For agricultural areas, data points predominantly clustered near the upper vertex of the ternary plot, indicating limitations in both vegetation and soil C storage due to conventional farming practices. Here, management strategies should prioritize soil-focused measures, specifically adopting conservation agriculture practices such as retaining crop residues as surface mulch, reducing tillage intensity by shifting toward no-till or strip-till systems, and integrating biochar and compost applications to bolster soil C storage [69,70].

Urban parks exhibited management needs concentrated around soil C storage, largely driven by factors unique to urban green spaces, such as soil compaction and limited organic inputs. Therefore, targeted interventions that alleviate compaction through periodic aeration combined with organic mulch or compost amendments, preserve litter layers, and multilayered plantings using deep-rooted shrub species can effectively enhance soil C storage while supporting vegetation C storage simultaneously [71,72]. In addition, enhancing connectivity by linking parks through green corridors and adding small complementary green spaces such as pocket parks and vegetated street nodes as stepping stones can increase effective patches and strengthen C storage and sequestration in urban parks [65].

In contrast, roadside vegetation displayed more complex, multi-dimensional poor C patterns, with points frequently appearing near the ternary plot's center, suggesting concurrent poor condition in all three UGCI components. Effective management strategies in roadside vegetation should therefore integrate diverse practices, including designing multi-layered roadside plantings that incorporate drought-tolerant understory species to increase vegetation C and their sustainability. Applying pruning regimes that maintain canopy vigor and sustained growth while avoiding excessive biomass removal, and mitigating soil sealing and compaction by enlarging planting strips and using permeable surfaces can be feasible to improve vegetation and soil C status.

Thus, our two-step decision-supporting framework using UGCI and ternary plots allows policymakers to precisely identify management priorities by land cover types and devise targeted strategies that align with local environmental and ecological characteristics, maximizing cost-effectiveness in urban C management.

### 3.4. Comparison between UGCI and an existing carbon management index

To evaluate whether the UGCI provides a more effective metric for urban C management than previously proposed index, we compared its statistical and spatial distributions with those of the Carbon Sequestration Potential Index (CSPI) developed by Pascual et al. (2020). The detailed methodology for calculating the CSPI is provided in Text S4 in S1 File. The data distribution of the UGCI was relatively closer to a normal distribution (Fig S3 in S1 File), supporting its higher discriminative capacity for assessing C conditions within heterogeneous urban environments.

Figs 7a and 7b illustrate the spatial distributions of UGCI and CSPI, respectively. Because the UGCI and CSPI were developed for different purposes, their spatial patterns across the urban landscape differed markedly. In the UGCI map, clear gradients were observed between forest edges and interiors within forest patches, whereas the CSPI did not capture such within-patch variations, showing near-zero values across forested areas. This is because the CSPI was designed to directly represent the potential only for afforestation management, thus assigning low potential to areas that are already forested. Unlike the CSPI, the UGCI captures both low-biomass areas requiring vegetation establishment and high-biomass forests with limited NEP or soil C condition, thereby identifying diverse C management needs across urban landscapes.

High CSPI values were mostly found in agricultural areas (Figs 7c and 7d), which reflects their low aboveground C density. However, because agricultural systems are managed to maintain intentionally low aboveground biomass compared to forests, the CSPI—constructed solely from vegetation-related components—has limited capability in evaluating the C status of non-forest green spaces. Considering the diverse land-cover types in urban green spaces, the UGCI's integrated vegetation and soil scheme offers a more comprehensive representation of urban C storage and sequestration potential.

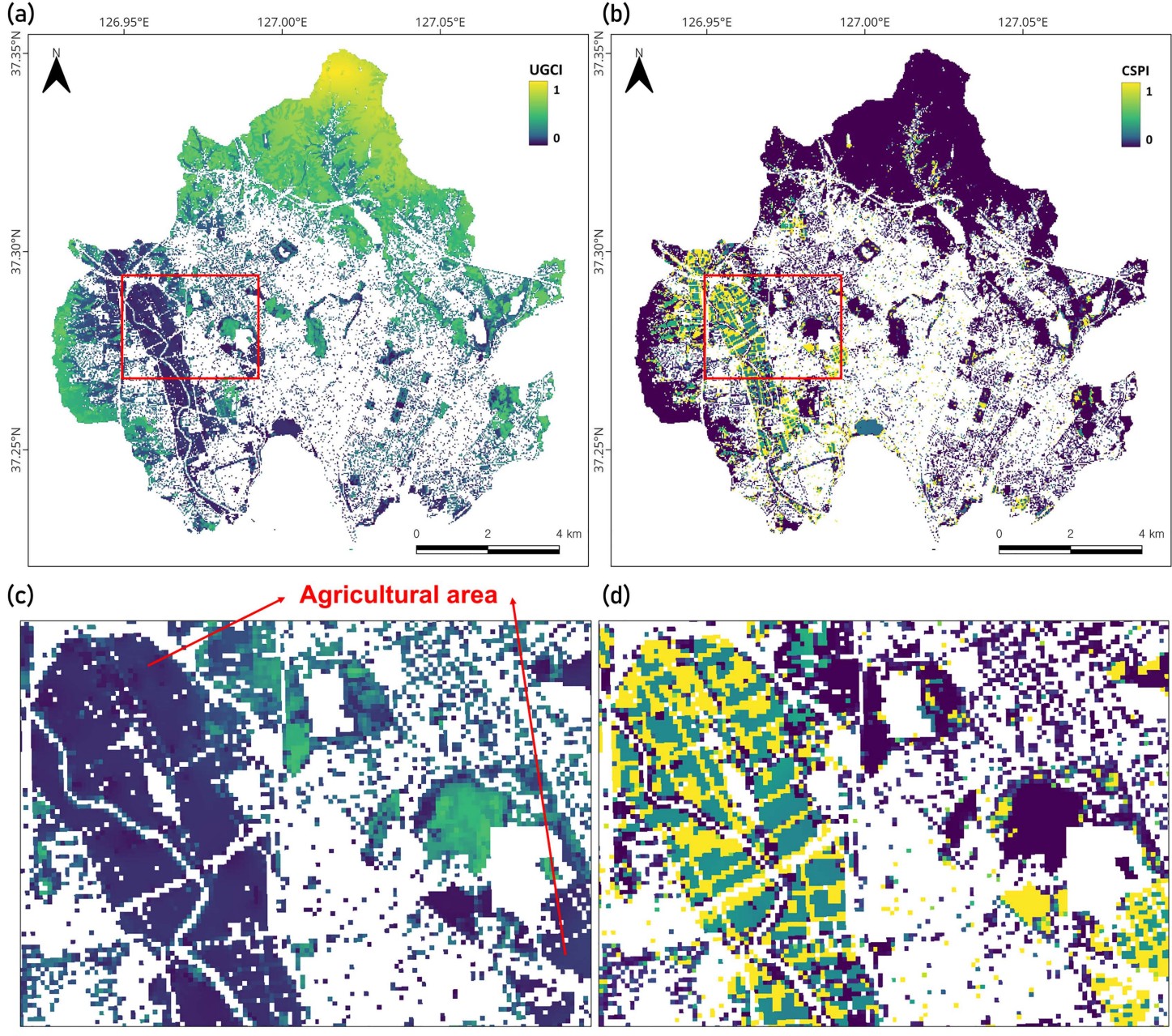

**Fig 7. Comparison between the Urban Green Carbon Index (UGCI) and the Carbon Sequestration Potential Index (CSPI).** Spatial distributions of (a) UGCI and (b) CSPI across the study area are shown. The red box marks the region of interest, for which detailed views are provided in (c) UGCI, and (d) CSPI.

## 3.5. Limitation and future study

This study had several clear limitations that can be addressed in future research. First, the UGCI was currently focused on C enhancement and did not integrate policy implementation pathways or cost-benefit considerations. Because its primary role was to provide a scientifically grounded, spatially explicit diagnosis of low-carbon areas and management directions,

economic assessments were intentionally kept external to the index. Future studies could link the UGCI with additional decision layers, such as cost-effectiveness, marginal abatement cost, life-cycle assessment, or budgeting, through multi-criteria decision analysis. This would enhance the applicability of the index while preserving flexibility to local financial capacities and policy priorities.

Second, the UGCI framework did not explicitly account for ecological interactions or nonlinear feedback among its constituent components. Once these interrelationships are quantitatively understood, incorporating them into the index could improve its ability to represent complex ecosystem processes and capture spatiotemporal variability in C dynamics.

Lastly, the reliability of UGCI analysis depends on the availability of fine-scale datasets and validation through field observations. Because grid-based analysis involves resampling multiple datasets, collecting the finest original data is critical. Although this study performed validation using available field data, the limiting number of observation system (e.g., eddy covariance tower) may introduce biases related to seasonality, vegetation composition, and extreme climate events. To reduce uncertainty in the net C uptake of UGCI, future work should expand validation efforts by establishing additional micrometeorological measurements and chamber-based flux observations across distinct urban green space types (e.g., pocket parks, roadside plants, and green roofs), enabling direct calibration and validation of the model estimations. Continued development of high-resolution datasets and the expansion of observation networks across diverse land-cover types will be essential for improving the accuracy and generalization of UGCI-based assessments.

## 4. Conclusions

In this study, we proposed a framework that integrates the UGCI with a multi-dimensional analytical approach to connect spatial diagnosis with management suggestion. Not limited to quantifying vegetation and soil C stocks, the framework incorporates sequestration potential and pinpoints how urban heterogeneity, including edge effects, impervious adjacency, and fragmentation, creates within-site differences in C storage and uptake that can guide targeted management. By translating these within-patch contrasts into spatially explicit, actionable signals, the framework helps practitioners prioritize and tailor interventions at the scale of individual patches. The UGCI, coupled with ternary-plot analysis, allows users to disentangle the relative contributions and limitations of vegetation, soil, and net C uptake, thereby identifying where and how each component can be strengthened. This capability is particularly valuable in complex urban matrices where biophysical heterogeneity and land-use interactions limit the effectiveness of one-size-fits-all approaches. Building on this framework, future work should focus on improving robustness through higher-resolution datasets and expanded field validation. Applying and comparing the framework across multiple cities will test its transferability and inform context-specific C management strategies.

## Supporting information

**S1 File. Supporting information.**
(DOCX)

## Author contributions

**Conceptualization:** Inhye Seo, Gayoung Yoo.

**Funding acquisition:** Sujong Jeong.

**Investigation:** Inhye Seo.

**Methodology:** Inhye Seo.

**Resources:** Bokyung Son, Yeonsu Lee, Jongho Kim.

**Supervision:** Gayoung Yoo.

**Validation:** Bokyung Son, Yeonsu Lee, Jongho Kim.

**Visualization:** Inhye Seo.

**Writing – original draft:** Inhye Seo, Bokyung Son, Jongho Kim.

**Writing – review & editing:** Junge Hyun, Jungho Im, Sujong Jeong, Gayoung Yoo.

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
