## [Decision Letter · Decision Letter 0]

24 Sep 2025

PONE-D-25-21587A Comprehensive Decision-making Framework for Prioritizing Carbon Management in Complex Urban LandscapePLOS ONE

Dear Dr. Yoo,

Thank you for submitting your manuscript to PLOS ONE. After careful consideration, we feel that it has merit but does not fully meet PLOS ONE’s publication criteria as it currently stands. Therefore, we invite you to submit a revised version of the manuscript that addresses the points raised during the review process.

We look forward to receiving your revised manuscript.

Kind regards,

Tianheng Shu, PhD

Academic Editor

PLOS ONE

Journal Requirements:

“This work was supported by Korea Environment Industry & Technology Institute (KEITI) through "Climate Change R&D Project for New Climate Regime.", funded by Korea Ministry of Environment (MOE) (2022003560006)”

5. We note that Figure 1, 2, 3 and 4 in your submission contain map images which may be copyrighted. All PLOS content is published under the Creative Commons Attribution License (CC BY 4.0), which means that the manuscript, images, and Supporting Information files will be freely available online, and any third party is permitted to access, download, copy, distribute, and use these materials in any way, even commercially, with proper attribution. For these reasons, we cannot publish previously copyrighted maps or satellite images created using proprietary data, such as Google software (Google Maps, Street View, and Earth). For more information, see our copyright guidelines: http://journals.plos.org/plosone/s/licenses-and-copyright.

1. You may seek permission from the original copyright holder of Figure 1, 2, 3 and 4 to publish the content specifically under the CC BY 4.0 license.

Reviewers' comments:

Reviewer's Responses to Questions

**Comments to the Author**

1. Is the manuscript technically sound, and do the data support the conclusions?

Reviewer #1: Yes

Reviewer #2: No

Reviewer #3: Partly

Reviewer #4: Partly

Reviewer #5: Yes

2. Has the statistical analysis been performed appropriately and rigorously? 

Reviewer #1: Yes

Reviewer #2: No

Reviewer #3: No

Reviewer #4: No

Reviewer #5: No

3. Have the authors made all data underlying the findings in their manuscript fully available?

Reviewer #1: Yes

Reviewer #2: Yes

Reviewer #3: Yes

Reviewer #4: No

Reviewer #5: Yes

4. Is the manuscript presented in an intelligible fashion and written in standard English?

Reviewer #1: No

Reviewer #2: Yes

Reviewer #3: Yes

Reviewer #4: Yes

Reviewer #5: Yes

5. Review Comments to the Author

Reviewer #1: This paper presented a spatial decision-making framework for the urban carbon sequestration management based on a novel Carbon Management Prioritizing Index (CMPI) and the ternary plots. The CMPI integrates the spatial data on vegetation C storage, soil C storage, net C uptake, and soil C storage potential and complete the diagnosis procedure and the ternary plots implements site-specific C management strategies. This framework was applied to a case study in Suwon, South Korea. Generally, this research is interesting and the framework can be applicable to urban C management. The writing-up of this manuscript is generally acceptable, but some clarifications and improvements are still necessary. My particular comments are as follows.

1. What is the minimum grid size for a landscape simulation? This one should be discussed in details because it concerns the balance between computation load and accuracy.

2. All equations should have their number.

3. Is whether ‘Section 2.2 Case Study’ here suitable or not? Is the framework associated with case study? My understanding is that Case study only demonstrates the application of this framework.

4. English and presentations should be significantly improved for clear and correct statements or expressions

Reviewer #2: This study presents a spatially explicit framework for diagnosing and prioritizing urban carbon management using a newly developed Carbon Management Prioritizing Index (CMPI) and ternary plots. Overall, the study is timely and policy-relevant. I particularly appreciate the integration of both vegetation and soil carbon components and the use of high-resolution data in a highly urbanized context. However, several issues need to be addressed to improve the scientific rigor, policy applicability, and clarity of the manuscript. I recommend major revision before further consideration for publication.

1. Silt and clay content is an intrinsic parent-material property that cannot be manipulated by urban green-space management on policy-relevant timescales. Thus, I think that including it in a “management prioritisation index” (CMPI) is conceptually invalid. Moreover, the use of silt and clay content alone as a proxy for soil carbon potential is overly simplistic and may overlook key edaphic factors (e.g., pH, bulk density, microbial activity). In addition, due to its high spatial heterogeneity at the fine scale, silt and clay content receives the highest weight (>0.4), while vegetation C stock, the only component directly manageable by urban forestry, is down-weighted. The outcome is a mathematically tractable but ecologically meaningless index.

2. I am confused that high-biomass forests receive low CMPI scores and are consequently ranked as “low priority”, whereas low-biomass roadside plantings receive high CMPI scores. The authors implicitly assume that marginal C gains are automatically larger where current stocks are low, but supply no cost–benefit or marginal abatement cost curve to support this assumption. In fact, mature forests often provide the largest, cheapest avoided emissions simply by conserving existing carbon; ignoring this principle undermines the entire prioritisation logic.

3. A unit loss in vegetation C is compensated by an equivalent gain in soil C or NEP. In reality vegetation removal often accelerates soil C loss, producing positive feedbacks. The linear model ignores the inter-process coupling and nonlinear thresholds and synergies, which is a primary error in ecological modeling.

4. Soil C data are taken from the 1 km Global Soil Organic Carbon map (GSOC v1.6) and resampled to 30 m with bilinear interpolation. Interpolation cannot create new information; the “30 m” soil layer still carries 1 km effective support. The validation R2 = 0.84 derives from 121 national soil samples that were never designed to test 30 m spatial accuracy. Using this as evidence of high-resolution reliability is misleading.

5. GPP and Reco models are calibrated with Mt. Taehwa eddy-covariance data (a peri-natural forest on rugged terrain 400 m a.s.l.) and applied to inner-city grids where surface energy balance, aerosol load, irrigation, and heat-island effects differ drastically. No site-specific validation is provided for Suwon’s urban land covers. Systematic over-estimation of NEP is highly probable.

6. Roadside plots cluster near the centre of the ternary diagram and are declared “multi-dimensionally degraded”. Given their young age, small size and engineered substrates these plots are expected to start with low biomass; without a temporal baseline the authors cannot demonstrate a decline (degradation). The interpretation is therefore a logical fallacy.

7. Forest edges exhibit 34 % higher CMPI than interiors (p < 0.001). The authors immediately prescribe “buffer zones”, yet provide no evidence that the CMPI difference corresponds to measurable biomass loss or soil C reduction.

8. Urban parks show a weak negative correlation (r = –0.27) between patch size and CMPI. The authors conclude that “smaller patches need intervention”. However, small parks are often sport fields or playgrounds with intentionally low biomass; again the paper confers a management necessity onto a land-use pattern that is by design.

9. The manuscript closes with generic advice: “create buffer zones”, “reduce soil compaction”, “apply multi-layer planting”. No cost estimates, no trade-off analyses, no references to municipal codes or land-opportunity prices (USD > 1 000 m2 in Suwon) are provided. Such recommendations are not actionable and fall well short of PLOS ONE’s requirement that papers offer “evidence-based, practical solutions”.

Reviewer #3: The results and discussion section is too weak to support the final conclusion in this manuscript. in contrast, the author put too many details on the materials/methods section, which make the structure of this manuscript is unbalanced. Also, all figures are in a limited quality. I don't think the current manuscript meet the standard of the submitted journal.

Reviewer #4: This manuscript presents an interesting framework for a Carbon Management Priority Index (CMPI) that integrates vegetation, soil, and productivity indicators at fine resolution for urban carbon management. Several issues need to be addressed:

1. The manuscript relies on multi-resolution datasets that are all resampled to 30 m, yet the methods for resampling are insufficiently explained. Please clarify exactly how each dataset was resampled, justify the choices, and provide an assessment of how resampling errors may affect CMPI outputs.

2. The refinement of land-use data from 1 m to 0.25 m resolution also needs stronger justification (Line 201-210). Since the final analysis is conducted at 30 m, it is unclear whether this refinement adds meaningful information beyond additional processing. Please provide quantitative evidence that this improves classification accuracy and ultimately influences CMPI estimates.

3. In Section 2, the roadside vegetation correction model, the U-Net classification, and the digital soil mapping are described only briefly, without sufficient information on model type, input features, training and validation procedures, hyperparameters, or performance metrics. These details are critical for reproducibility and should be provided, at least in supplementary materials.

4. The discussion does not sufficiently situate CMPI in relation to existing indices and prior studies. A more explicit comparison with other urban carbon assessment methods would help demonstrate what CMPI adds, both in terms of input variables, weighting strategies, and management relevance. Even a qualitative comparison, if quantitative replication is not feasible, would make the contribution more convincing.

5. Line428-429, the correlation reported between patch size and CMPI (r = –0.27, p < 0.001) is weak. This does not justify the strong claim that smaller patches “generally” had higher CMPI values.

Reviewer #5: Dear Authors,

I have read carefully the manuscript entitled "A Comprehensive Decision-Making Framework for Prioritizing Carbon Management in Complex Urban Landscape" submitted by Seo et al. to PLOS One.

This manuscript seeks to introduce a novel decision-making framework designed to evaluate existing carbon storage and sequestration and to identify areas exhibiting deteriorated carbon dynamics.

Based on my review, I find that this manuscript addresses a highly relevant topic, given the urgent need for carbon neutrality strategies in urban areas. The proposed framework offers a practical and scientifically sound tool to support public policies for carbon management in urban landscapes. However, several issues require careful attention before the manuscript can be considered for publication. The key points are outlined below:

1) MATERIALS AND METHODS

1.1 The authors use EVI, LSWI, PAR, Tair, and VPD, but they do not discuss why these indices were chosen instead of others (e.g., NDVI, drought index, soil moisture).

1.2 In the Reco equation, the parameters a, b, and c are introduced, but their values, fitting ranges, or calibration methods are not provided. This omission limits the reproducibility of the study.

1.3 The reported R◯2 = 0.76 indicates a reasonably good model fit, but it is not particularly strong. The authors do not discuss the limitations of the model, such as potential biases related to seasonality, differences among vegetation types, or the influence of extreme events.

1.4 Although the authors indicate the units for GPP, they do not do the same for Reco in the corresponding formula.

1.5 It is only implied, but not clearly stated, whether the calculation is performed on a daily or hourly basis.

1.6 Validation using only one flux tower (Mt. Taehwa) may introduce bias. Consequently, the model may not generalize well across different land-cover types.

1.7 The authors defined 30 m as the threshold, but they did not justify this choice. It is unclear whether this value was based on previous literature, the resolution of the datasets, or a methodological decision.

2) RESULTS AND DISCUSSION

2.1 Figure 3c shows NEP ranging from –0.69 to 16.95 t C ha-1 yr-1. Could the authors clarify the negative value?

2.2 The authors state that CMPI is “more effective” than previous approaches, but the comparison appears mainly descriptive. Could the authors provide a quantitative comparison or benchmarking against existing indices?

2.3 How do the authors assess the propagation of uncertainties from the GSOC data and the GPP/Reco modeling into the CMPI values?

2.4 The results represent a specific time period, but urban carbon dynamics can vary under extreme climate events. Do the authors plan to validate the CMPI using longer time series or under climate change scenarios?

6. PLOS authors have the option to publish the peer review history of their article (what does this mean?). If published, this will include your full peer review and any attached files.

Reviewer #1: No

Reviewer #2: **Yes:** Hanqing Wu

Reviewer #3: No

Reviewer #4: No

Reviewer #5: No

---

## [Author Response · Author response to Decision Letter 1]

4 Jan 2026

Response to Reviewers [PONE-D-25-21587]

Dear Dr. Tianheng Shu, Academic editor:

Thank you for the opportunity to revise our manuscript, ID: PONE-D-25-21587, titled “A Comprehensive Decision-making Framework for Prioritizing Carbon Management in Complex Urban Landscape .” We are grateful to you and the esteemed reviewers for the time and effort dedicated to providing insightful and constructive feedback. The comments have been invaluable in helping us strengthen the manuscript significantly.

We have thoroughly addressed all the points raised by the reviewers and have revised the manuscript accordingly.

The main changes incorporated into the revised manuscript are summarized below:

Conceptual refinement of the index: Following a critical suggestion from Reviewer #2, we have renamed the "Carbon Management Prioritizing Index (CMPI)" to the "Urban Green Carbon Index (UGCI)." This change more accurately reflects the index's function as a diagnostic tool for assessing carbon status, distinct from a full cost-benefit or opportunity-cost analysis, thereby clarifying the manuscript's core contribution. Accordingly, we changed the title of the manuscript from ‘A Comprehensive Decision-making Framework for Prioritizing Carbon Management in Complex Urban Landscape’ to ‘The urban green carbon index (UGCI): A spatial framework for suggesting urban carbon management.’

Strengthened methodological justification: We have provided more detailed explanations for key methodological decisions. This includes a clearer rationale for selecting the 30 m grid size, justification for the choice of input variables for our GPP/Reco models (as suggested by Reviewer #5), and a more transparent description of the data resampling and processing techniques (as requested by Reviewer #4).

Enhanced validation and contextualization: To address comments regarding the validation of our models and the novelty of our index (Reviewers #2, #4, and #5), we have expanded our discussion. We now include a direct quantitative comparison of the UGCI with an existing metric, the Carbon Sequestration Potential Index (CSPI), to better situate our work within the current literature and highlight its unique contributions to urban carbon management.

Refinement of interpretations and claims: Based on the feedback concerning potential logical fallacies (Reviewers #2 and #4), we have carefully revised our interpretations. For instance, we replaced ambiguous terms like "degraded" with more precise descriptions such as "under poor condition" and have moderated our claims where correlations were weak, ensuring our conclusions are robustly supported by the evidence presented.

Improved reproducibility and clarity: In response to requests for greater detail (Reviewers #4 and #5), we have added specific parameters for equations, elaborated on model validation procedures, and moved detailed methodological descriptions to the supplementary materials to ensure clarity and reproducibility without disrupting the flow of the main text. The entire manuscript has also undergone a thorough proofread to improve language and readability.

Here, we provide responses following the comments by the reviewers.

Reviewer #1:

This paper presented a spatial decision-making framework for the urban carbon sequestration management based on a novel Carbon Management Prioritizing Index (CMPI) and the ternary plots. The CMPI integrates the spatial data on vegetation C storage, soil C storage, net C uptake, and soil C storage potential and complete the diagnosis procedure and the ternary plots implements site-specific C management strategies. This framework was applied to a case study in Suwon, South Korea. Generally, this research is interesting and the framework can be applicable to urban C management. The writing-up of this manuscript is generally acceptable, but some clarifications and improvements are still necessary. My particular comments are as follows.

Response: Thank you for your positive and encouraging review. We sincerely appreciate the thoughtful and constructive feedback. Your suggestions have significantly strengthened the manuscript, and we have revised the article to incorporate them. My detailed responses to each of your comments are outlined below.

1. What is the minimum grid size for a landscape simulation? This one should be discussed in details because it concerns the balance between computation load and accuracy.

Response: We appreciate this valuable comment. In our case study, the spatial resolution of the four components constituting the framework was unified to 30 m. This resolution was chosen by balancing the minimum available spatial resolution of the input datasets and the spatial granularity necessary for urban-scale analysis. Although a 1 m land cover map was available for assessing vegetation C storage, the other components lacked data at such a fine scale. Soil C storage was derived from the GSOCmap at 1 km resolution, and net C uptake component and soil silt and clay content were modeled at 250 m and 30 m resolution, respectively, reflecting the finest spatial resolutions of the available input dataset. Table 1 describes detailed information on input datasets. We have added this explanation to Section 2.2 Case study to clarify the rationale for selecting a 30 m grid.

Revised Text (Lines 190-193): In this case study, the data collection was conducted for the year 2020. All spatial data layers were resampled to a 30 m resolution, balancing the finest available resolution of input datasets with the spatial granularity required for urban-scale analysis.

2. All equations should have their number.

Response: Thanks for pointing it out. We have revised the manuscript so that all equations are now numbered consistently and can be clearly referenced in the text (Lines 124, 131, 143, 146, 150, 167, 228, 246, 288-289, 309).

3. Is whether ‘Section 2.2 Case Study’ here suitable or not? Is the framework associated with case study? My understanding is that Case study only demonstrates the application of this framework.

Response: We appreciate the reviewer’s constructive comment. The case study section was included not as an independent example, but as an essential process to demonstrate the applicability of the proposed framework in an actual urban environment. The framework itself is conceptual and methodological, and its practical relevance can only be evaluated through real-world implementation. Therefore, the case study serves as a proof of concept, showing how the framework operates under real urban conditions, including data acquisition and generation, possible analysis, and interpretation. For this reason, we maintained it as a dedicated section. To clarify the role of the case study within this research, we have added the purpose of implementing this case study at the beginning of Section 2.2 Case Study.

Original Text (Lines 187-190): To validate the effectiveness of the proposed framework, we applied it to Suwon, a representative metropolitan city in South Korea. Suwon (37°17'28" N, 127°0'32" E) is the third most densely populated city in the country, with a population density of 9,884 persons km⁻² as of 2024 (Korean Ministry of the Interior and Safety).

Revised Text (Lines 181-185): Through the case study, we demonstrate the applicability of the proposed framework and evaluate the practical availability of UGCI in an actual urban environment. Our study site was Suwon (37°17'28" N, 127°00'32" E), a representative metropolitan city in South Korea with a population density of 9,884 persons km⁻² as of 2024 (Korean Ministry of the Interior and Safety).

4. English and presentations should be significantly improved for clear and correct statements or expressions

Response: We sincerely appreciate the reviewer’s feedback regarding language clarity. The entire manuscript has been thoroughly revised to improve readability and precision in expression. All sentences were carefully re-checked using both professional editing tools and manual proofreading to ensure grammatical accuracy and clarity.

Reviewer #2:

This study presents a spatially explicit framework for diagnosing and prioritizing urban carbon management using a newly developed Carbon Management Prioritizing Index (CMPI) and ternary plots. Overall, the study is timely and policy-relevant. I particularly appreciate the integration of both vegetation and soil carbon components and the use of high-resolution data in a highly urbanized context. However, several issues need to be addressed to improve the scientific rigor, policy applicability, and clarity of the manuscript. I recommend major revision before further consideration for publication.

Response: Thank you for your positive and encouraging feedback on the manuscript. We sincerely appreciate the time you took to provide such thoughtful and constructive suggestions. The manuscript has been revised to incorporate these changes, which we believe have significantly improved the article. Please find our detailed responses to each of your comments outlined below.

1. Silt and clay content is an intrinsic parent-material property that cannot be manipulated by urban green-space management on policy-relevant timescales. Thus, I think that including it in a “management prioritisation index” (CMPI) is conceptually invalid. Moreover, the use of silt and clay content alone as a proxy for soil carbon potential is overly simplistic and may overlook key edaphic factors (e.g., pH, bulk density, microbial activity). In addition, due to its high spatial heterogeneity at the fine scale, silt and clay content receives the highest weight (>0.4), while vegetation C stock, the only component directly manageable by urban forestry, is down-weighted. The outcome is a mathematically tractable but ecologically meaningless index.

Response: We appreciate the reviewer’s valuable and constructive comments. We fully agree that soil texture is an intrinsic property that is not easily altered and may not always serve as a direct management lever in urban green policy. However, it is important to note that our study encompasses urban green spaces, which were not represented by single and undisturbed ecosystems such as natural forests. Most urban soils are constructed or restored, typically consisting of sandy materials used as fill. Under such conditions, the composition of silt and clay can be modified or selected during construction and restoration processes.

We also acknowledge that silt and clay content alone cannot represent all edaphic conditions. Nevertheless, soil texture is known as a primary parameter governing key physical and biological properties such as bulk density, aeration, and microbial habitat, which are fundamental to C protection and turnover. Because pH does not exhibit a direct relationship with C status, we did not include it in our index. These points have been incorporated into Section 2.2.4 Soil C storage potential: Silt and clay content.

Original Text (Lines 308-310): Because the soil silt and clay content critically influences soil C storage potential by enhancing soil’s capacity to stabilize C [39,40], we considered it as an indicator of soil C storage potential.

Revised Text (Lines 321-327): Given that soil silt and clay content play a critical role in stabilizing C [39,40], it was used as an indicator of soil C storage potential. While silt and clay content is intrinsic and relatively stable in natural soils, urban soils—particularly those in parks and roadside vegetation—are often reconstructed and sand-dominated, allowing urban soil texture to be modified during construction. Other edaphic factors, such as pH and microbial activity, were not incorporated because their direct relationships with soil C storage potential remain uncertain.

In addition, the weight for silt and clay content component was the lowest of 0.08, we have added the exact number in Section 3.3 Site-specific C management strategies and policy implications.

Original Text (Lines 443-445): Since soil silt and clay content exhibited minimal variability, it was excluded from the ternary plots, focusing on vegetation C storage, soil C storage, and net C uptake.

Revised Text (Lines 459-461): Among the components of the UGCI, the soil silt and clay content had the lowest weighting factor (0.08), indicating the smallest contribution to the overall variation of the index; therefore, it was excluded from the ternary plot analysis.

2. I am confused that high-biomass forests receive low CMPI scores and are consequently ranked as “low priority”, whereas low-biomass roadside plantings receive high CMPI scores. The authors implicitly assume that marginal C gains are automatically larger where current stocks are low, but supply no cost–benefit or marginal abatement cost curve to support this assumption. In fact, mature forests often provide the largest, cheapest avoided emissions simply by conserving existing carbon; ignoring this principle undermines the entire prioritisation logic.

Response: We appreciate your critical view. We recognize that the term ‘Carbon Management Prioritizing Index’ could be misleading, since management priority may imply the inclusion of cost-benefit or opportunity cost analyses. However, our research intention was to provide spatially explicit information on vegetation and soil C status prior to determining C management priorities. Therefore, we revised the name of index from CMPI to urban green carbon index (UGCI) throughout the manuscript. As we acknowledge the limitation of not incorporating policy implementation pathways and cost–benefit analysis, we included this limitation in Section 2.1 where UGCI was first introduced and in Section 3.5. Limitation and future study.

Original Text (Lines 106-108): This approach ensures that proposed management actions are both locally appropriate and practically effective for enhancing urban C sequestration capacity.

Revised Text (Lines 105-107): The proposed strategies do not incorporate cost-benefit or opportunity-cost analyses; these limitations and the corresponding directions for future research are discussed separately in the discussion section.

(Lines 534 - 542) This study had several clear limitations that can be addressed in future research. First, the UGCI was currently focused on C enhancement and did not integrate policy implementation pathways or cost-benefit considerations. Because its primary role was to provide a scientifically grounded, spatially explicit diagnosis of low-carbon areas and management directions, economic assessments were intentionally kept external to the index. Future studies could link the UGCI with additional decision layers, such as cost-effectiveness, marginal abatement cost, life-cycle assessment, or budgeting, through multi-criteria decision analysis. This would enhance the applicability of the index while preserving flexibility to local financial capacities and policy priorities.

The assumption pointed out by the reviewer that areas with low carbon stocks may exhibit high marginal carbon gains is a concept commonly applied in the context of urban and anthropogenically managed green spaces. For instance, Pascual et al. (2020) developed the Carbon Sequestration Potential Index (CSPI) to identify priority areas for afforestation and reforestation, assigning higher index values to regions with lower forest cover or lower photosynthetic efficiency per unit biomass. Georgiou et al. (2022) demonstrated that soils with a greater deficit from their carbon-saturation capacity accumulated carbon more rapidly through management interventions than soils approaching saturation. These studies collectively support the rationale underlying our assumption. Nevertheless, we fully agree with the reviewer that the conservation of mature forests is a critical management strategy from the perspective of avoided carbon emissions. However, low index values do not necessarily indicate that management is unnecessary; rather, it can lead to conservation-oriented management strategies. We have added this clarification to the Section 3.3. Site-specific C management strategies and policy implications.

Original Text (Lines 442-445): Figure 7 presents ternary plots illustrating grids classified as ‘high’ and ‘extremel

---

## [Decision Letter · Decision Letter 1]

27 Jan 2026

PONE-D-25-21587R1The urban green carbon index (UGCI): A spatial framework for suggesting urban carbon managementPLOS One

Dear Dr. Yoo,

Thank you for submitting your manuscript to PLOS ONE. After careful consideration, we feel that it has merit but does not fully meet PLOS ONE’s publication criteria as it currently stands. Therefore, we invite you to submit a revised version of the manuscript that addresses the points raised during the review process.

We look forward to receiving your revised manuscript.

Kind regards,

Tianheng Shu, PhD

Academic Editor

PLOS One

Journal Requirements:

Additional Editor Comments (if provided):

The reviewers have further raised some minor concerns. Please make some changes to your paper accordingly.

Reviewers' comments:

Reviewer's Responses to Questions

**Comments to the Author**

1. If the authors have adequately addressed your comments raised in a previous round of review and you feel that this manuscript is now acceptable for publication, you may indicate that here to bypass the “Comments to the Author” section, enter your conflict of interest statement in the “Confidential to Editor” section, and submit your "Accept" recommendation.

Reviewer #1: All comments have been addressed

Reviewer #2: All comments have been addressed

Reviewer #3: All comments have been addressed

Reviewer #4: All comments have been addressed

Reviewer #5: All comments have been addressed

2. Is the manuscript technically sound, and do the data support the conclusions?

Reviewer #1: Yes

Reviewer #2: Yes

Reviewer #3: Yes

Reviewer #4: Yes

Reviewer #5: Yes

3. Has the statistical analysis been performed appropriately and rigorously? 

Reviewer #1: Yes

Reviewer #2: Yes

Reviewer #3: Yes

Reviewer #4: Yes

Reviewer #5: Yes

4. Have the authors made all data underlying the findings in their manuscript fully available?

Reviewer #1: Yes

Reviewer #2: Yes

Reviewer #3: Yes

Reviewer #4: Yes

Reviewer #5: Yes

5. Is the manuscript presented in an intelligible fashion and written in standard English?

Reviewer #1: Yes

Reviewer #2: Yes

Reviewer #3: Yes

Reviewer #4: Yes

Reviewer #5: Yes

6. Review Comments to the Author

Reviewer #1: The authors have well addressed our comments and improved the quality of this manuscript. I have no more technical comments.

Reviewer #2: This study presents a novel, spatially explicit decision-support framework that integrates vegetation carbon storage, soil carbon storage, net carbon uptake, and soil carbon storage potential into a composite “Urban Green Carbon Index (UGCI)”. Coupled with ternary plot analysis, the framework provides a diagnostic tool and a basis for prescribing site-specific management strategies for urban carbon sequestration. The research addresses a scientifically significant and policy-relevant topic. The methodological approach is robust, combining high-resolution remote sensing, modeling, and machine learning techniques. The case study in Suwon, South Korea, is well-designed, and the results are presented and analyzed thoroughly. The authors have undertaken extensive and thoughtful revisions in response to the reviewers’ comments. Key improvements include the conceptual refinement of the index (renaming from CMPI to UGCI), strengthened methodological justifications, enhanced validation and contextualization through comparison with an existing index, more nuanced interpretation of results, and clearer acknowledgment of limitations. These revisions have significantly strengthened the manuscript's clarity, rigor, and impact.

While the manuscript is of high quality, the following minor points could further enhance it:

1. In Section 2.2.4 or the Discussion/Limitations, consider adding a sentence explicitly stating that soil texture is used as a feasible proxy indicator for physical carbon stabilization potential, acknowledging that future iterations of UGCI could incorporate more direct measures of soil carbon saturation deficit or stabilization capacity as such data become available at relevant scales.

2.The authors correctly acknowledge the limitation of using a single flux tower for model validation and extrapolation to diverse urban interiors. In Section 3.5, this point could be slightly expanded. It could be suggested that future work aims to establish more micrometeorological or chamber-based flux measurements across distinct urban green space types (e.g., pocket parks, green roofs) to directly calibrate/validate carbon flux models for the specific urban matrix, thereby reducing uncertainty in the NEP component of UGCI.

3. In Section 3.3, when listing strategies like “reducing soil compaction” or “multi-layered planting”, you could add a brief illustrative example without delving into full cost analysis. For instance: “...such as using organic mulch amendments and planting deep-rooted species to alleviate compaction in urban parks”, or “...prioritizing drought-tolerant understory species in multi-layered roadside plantings”. This adds specificity while maintaining the focus on science-driven diagnosis.

This is a well-executed study with strong conceptual and methodological contributions to the field of urban carbon management. The authors have successfully addressed the substantive concerns raised during review. The manuscript is now suitable for publication, pending incorporation of the minor, constructive suggestions outlined above.

Reviewer #3: The authors addressed most of the main comments from previous reviewers. I believe it makes the manuscript much better and is suitable for getting a publishment.

Reviewer #4: The authors have responded adequately and revised the manuscript accordingly. I have no further major concerns.

Reviewer #5: Dear Authors,

I have carefully read again the manuscript entitled "The Urban Green Carbon Index (UGCI): a spatial framework for suggesting urban carbon management" submitted by Seo et al. to PLOS One.

This manuscript seeks to introduce a novel decision-making framework designed to evaluate existing carbon storage and sequestration and to identify areas exhibiting deteriorated carbon dynamics.

Based on my review, I am satisfied with the authors’ responses to my comments.

7. PLOS authors have the option to publish the peer review history of their article (what does this mean?). If published, this will include your full peer review and any attached files.

Reviewer #1: No

Reviewer #2: **Yes:** Hanqing Wu

Reviewer #3: No

Reviewer #4: No

Reviewer #5: No

---

## [Author Response · Author response to Decision Letter 2]

24 Mar 2026

Dear Dr. Tianheng Shu, Academic editor:

Thank you for the opportunity to revise our manuscript, ID: PONE-D-25-21587R1, titled “The urban green carbon index (UGCI): A spatial framework for suggesting urban carbon management.” We sincerely appreciate your guidance and the reviewers’ thoughtful comments throughout the review process. We have carefully addressed the additional comments from the reviewer and refined the manuscript accordingly.

The main changes incorporated into the revised manuscript are summarized below:

• Clarified the use of soil texture as a proxy indicator: In response to the suggestion from Reviewer #2, we explicitly state that soil silt and clay content is used as a feasible proxy for physical carbon stabilization potential. We also added a forward-looking note that UGCI could incorporate direct measures of soil carbon saturation deficit or stabilization capacity when spatially explicit datasets become available at relevant scales.

• Expanded limitations and future directions for NEP validation: We expanded Section 3.5 to more clearly acknowledge the limitation of validating NEP estimates using a single flux tower and the uncertainty associated with extrapolating to diverse urban land covers. We further suggested that future work should establish additional micrometeorological and/or chamber-based flux measurements across distinct urban green space types to directly calibrate/validate carbon flux model estimates, thereby reducing uncertainty in the NEP component of the UGCI.

• Added more specific illustrative examples to management strategies: According to the suggestion from Reviewer #2, we refined the management recommendations in Section 3.3 by integrating brief, concrete examples into key strategies. This improves practical clarity while maintaining the science-driven diagnostic focus without introducing a full cost analysis.

Here, we provide responses following the comments by the reviewers.

Reviewer #1:

The authors have well addressed our comments and improved the quality of this manuscript. I have no more technical comments.

Response: We sincerely thank the reviewer for the positive evaluation and for confirming that no further technical comments are needed. The reviewer’s previous suggestions were very helpful in improving the manuscript’s clarity and presentation. We appreciate the reviewer’s careful assessment, which has significantly contributed to enhancing the quality of the manuscript.

Reviewer #2:

This study presents a novel, spatially explicit decision-support framework that integrates vegetation carbon storage, soil carbon storage, net carbon uptake, and soil carbon storage potential into a composite “Urban Green Carbon Index (UGCI)”. Coupled with ternary plot analysis, the framework provides a diagnostic tool and a basis for prescribing site-specific management strategies for urban carbon sequestration. The research addresses a scientifically significant and policy-relevant topic. The methodological approach is robust, combining high-resolution remote sensing, modeling, and machine learning techniques. The case study in Suwon, South Korea, is well-designed, and the results are presented and analyzed thoroughly. The authors have undertaken extensive and thoughtful revisions in response to the reviewers’ comments. Key improvements include the conceptual refinement of the index (renaming from CMPI to UGCI), strengthened methodological justifications, enhanced validation and contextualization through comparison with an existing index, more nuanced interpretation of results, and clearer acknowledgment of limitations. These revisions have significantly strengthened the manuscript's clarity, rigor, and impact.

While the manuscript is of high quality, the following minor points could further enhance it:

Response: We sincerely thank the reviewer for the detailed and positive assessment of our revised manuscript. We are particularly grateful for the reviewer’s earlier in-depth critiques, which greatly improved the conceptual clarity, scientific rigor, and policy relevance of the study. The manuscript has been revised to incorporate the reviewer’s suggestions, which we believe have further improved the clarity of our article. Please find our detailed responses to each of your comments outlined below.

1. In Section 2.2.4 or the Discussion/Limitations, consider adding a sentence explicitly stating that soil texture is used as a feasible proxy indicator for physical carbon stabilization potential, acknowledging that future iterations of UGCI could incorporate more direct measures of soil carbon saturation deficit or stabilization capacity as such data become available at relevant scales.

Response: We appreciate the reviewer’s constructive suggestion. We have clarified that silt and clay content is used as a feasible proxy indicator of physical C stabilization potential, and could be updated with other direct measures if the spatially explicit data become available at relevant scales. These points have been incorporated into Section 2.2.4 Soil C storage potential: Silt and clay content.

Revised Text (Lines 323-330): Given that soil silt and clay content play a critical role in stabilizing C [39,40], we used silt and clay content as a feasible proxy indicator of physical C storage potential. If a more direct indicator for C storage potential, such as soil C saturation deficit or stabilization capacity, becomes available at a spatially relevant scale, the soil C storage potential component of UGCI could be updated to incorporate those direct metrics. While silt and clay content is intrinsic and relatively stable in natural soils, urban soils—particularly those in parks and roadside vegetation—are often reconstructed and sand-dominated, allowing urban soil texture to be modified during construction.

2.The authors correctly acknowledge the limitation of using a single flux tower for model validation and extrapolation to diverse urban interiors. In Section 3.5, this point could be slightly expanded. It could be suggested that future work aims to establish more micrometeorological or chamber-based flux measurements across distinct urban green space types (e.g., pocket parks, green roofs) to directly calibrate/validate carbon flux models for the specific urban matrix, thereby reducing uncertainty in the NEP component of UGCI.

Response: We thank the reviewer for this helpful recommendation. In Section 3.5 of our manuscript, we have suggested that uncertainty in the net C uptake of UGCI could be reduced by establishing additional observations across distinct urban green space types to directly calibrate/validate model estimates.

Revised Text (Lines 556-567): Lastly, the reliability of UGCI analysis depends on the availability of fine-scale datasets and validation through field observations. Because grid-based analysis involves resampling multiple datasets, collecting the finest original data is critical. Although this study performed validation using available field data, the limiting number of observation system (e.g., eddy covariance tower) may introduce biases related to seasonality, vegetation composition, and extreme climate events. To reduce uncertainty in the net C uptake of UGCI, future work should expand validation efforts by establishing additional micrometeorological measurements and chamber-based flux observations across distinct urban green space types (e.g., pocket parks, roadside plants, and green roofs), enabling direct calibration and validation of the model estimations. Continued development of high-resolution datasets and the expansion of observation networks across diverse land-cover types will be essential for improving the accuracy and generalization of UGCI-based assessments.

3. In Section 3.3, when listing strategies like “reducing soil compaction” or “multi-layered planting”, you could add a brief illustrative example without delving into full cost analysis. For instance: “...such as using organic mulch amendments and planting deep-rooted species to alleviate compaction in urban parks”, or “...prioritizing drought-tolerant understory species in multi-layered roadside plantings”. This adds specificity while maintaining the focus on science-driven diagnosis.

This is a well-executed study with strong conceptual and methodological contributions to the field of urban carbon management. The authors have successfully addressed the substantive concerns raised during review. The manuscript is now suitable for publication, pending incorporation of the minor, constructive suggestions outlined above.

Response: We appreciate the reviewer’s constructive suggestion. We agree that adding brief but illustrative details can improve the practical clarity of the strategies. Therefore, we revised Section 3.3 to embed concise examples as follows:

Revised Text (Lines 470-500): In forests, all grids classified as ‘low’ UGCI were spatially situated at edge areas, emphasizing the need for edge-focused management. These edge grids consistently exhibited the lowest scores for soil C storage, indicating that management efforts should primarily focus on enhancing soil C stability. Because forest-edge soils are more exposed to higher temperatures and direct sunlight, leading to accelerated decomposition [53,67], establishing shade-tolerant native understory shrubs/groundcovers and avoiding intensive leaf litter removal in edge areas can help minimize soil exposure and sustain soil moisture.

For agricultural areas, data points predominantly clustered near the upper vertex of the ternary plot, indicating limitations in both vegetation and soil C storage due to conventional farming practices. Here, management strategies should prioritize soil-focused measures, specifically adopting conservation agriculture practices such as retaining crop residues as surface mulch, reducing tillage intensity by shifting toward no-till or strip-till systems, and integrating biochar and compost applications to bolster soil C storage [71,72].

Urban parks exhibited management needs concentrated around soil C storage, largely driven by factors unique to urban green spaces, such as soil compaction and limited organic inputs. Therefore, targeted interventions that alleviate compaction through periodic aeration combined with organic mulch or compost amendments, preserve litter layers, and multilayered plantings using deep-rooted shrub species can effectively enhance soil C storage while supporting vegetation C storage simultaneously [73,74]. In addition, enhancing connectivity by linking parks through green corridors and adding small complementary green spaces such as pocket parks and vegetated street nodes as stepping stones can increase effective patches and strengthen C storage and sequestration in urban parks [64].

In contrast, roadside vegetation displayed more complex, multi-dimensional poor C patterns, with points frequently appearing near the ternary plot's center, suggesting concurrent poor condition in all three UGCI components. Effective management strategies in roadside vegetation should therefore integrate diverse practices, including designing multilayered roadside plantings that incorporate drought-tolerant understory species to increase vegetation C and their sustainability. Applying pruning regimes that maintain canopy vigor and sustained growth while avoiding excessive biomass removal, and mitigating soil sealing and compaction by enlarging planting strips and using permeable surfaces can be feasible to improve vegetation and soil C status.

Reviewer #3:

The authors addressed most of the main comments from previous reviewers. I believe it makes the manuscript much better and is suitable for getting a publishment.

Response: We greatly appreciate the reviewer’s thoughtful feedback and are pleased that the revised manuscript is now considered suitable for publication. The reviewer’s earlier comments were instrumental in improving the balance and strength of the manuscript. We thank the reviewer again for the constructive guidance that helped us substantially improve the manuscript.

Reviewer #4:

The authors have responded adequately and revised the manuscript accordingly. I have no further major concerns.

Response: We thank the reviewer for the encouraging comment and for confirming that there are no further major concerns. The reviewer’s earlier feedback was especially valuable for enhancing methodological transparency and reproducibility. We are grateful for the reviewer’s careful review, which significantly improved the rigor and clarity of the manuscript.

Reviewer #5:

Dear Authors,

I have carefully read again the manuscript entitled "The Urban Green Carbon Index (UGCI): a spatial framework for suggesting urban carbon management" submitted by Seo et al. to PLOS One.

This manuscript seeks to introduce a novel decision-making framework designed to evaluate existing carbon storage and sequestration and to identify areas exhibiting deteriorated carbon dynamics.

Based on my review, I am satisfied with the authors’ responses to my comments.

Response: We sincerely thank the reviewer for the careful re-evaluation of our revised manuscript and for noting that our responses have adequately addressed the previous concerns. The reviewer’s earlier comments were highly valuable for improving the methodological justification, reporting completeness, and discussion of limitations. We appreciate the reviewer’s constructive feedback, which helped strengthen the overall quality and robustness of the manuscript.

---

## [Decision Letter · Decision Letter 2]

7 Apr 2026

The urban green carbon index (UGCI): A spatial framework for suggesting urban carbon management

PONE-D-25-21587R2

Dear Dr. Yoo,

We’re pleased to inform you that your manuscript has been judged scientifically suitable for publication and will be formally accepted for publication once it meets all outstanding technical requirements.

Kind regards,

Tianheng Shu, PhD

Academic Editor

PLOS One

Additional Editor Comments (optional):

Reviewers' comments:

Reviewer's Responses to Questions

**Comments to the Author**

1. If the authors have adequately addressed your comments raised in a previous round of review and you feel that this manuscript is now acceptable for publication, you may indicate that here to bypass the “Comments to the Author” section, enter your conflict of interest statement in the “Confidential to Editor” section, and submit your "Accept" recommendation.

Reviewer #2: All comments have been addressed

2. Is the manuscript technically sound, and do the data support the conclusions?

Reviewer #2: Yes

3. Has the statistical analysis been performed appropriately and rigorously? 

Reviewer #2: Yes

4. Have the authors made all data underlying the findings in their manuscript fully available?

Reviewer #2: Yes

5. Is the manuscript presented in an intelligible fashion and written in standard English?

Reviewer #2: Yes

6. Review Comments to the Author

Reviewer #2: The authors have provided a thorough and well-structured response to the previous round of reviews, and the revisions made to the manuscript are both appropriate and clearly documented. All points raised by the reviewers have been addressed in a systematic manner. Specifically, the clarification regarding the use of silt and clay content as a feasible proxy for soil carbon storage potential is now explicitly stated, and the forward-looking statement about incorporating more direct indicators when data become available adds appropriate nuance. The expansion of the limitations section to acknowledge the uncertainty associated with using a single flux tower for NEP validation is well justified, and the proposed direction for future work involving additional micrometeorological or chamber-based flux measurements across diverse urban green space types strengthens the scientific rigor of the study. Furthermore, the integration of brief, concrete examples into the management strategies in Section 3.3 enhances practical clarity without shifting the focus from the science-driven diagnostic approach. The authors have demonstrated careful attention to detail, and the revised manuscript shows clear improvements in clarity, transparency, and methodological justification. I find that the revisions satisfactorily address the remaining minor concerns, and the manuscript is now suitable for publication.

7. PLOS authors have the option to publish the peer review history of their article (what does this mean?). If published, this will include your full peer review and any attached files.

Reviewer #2: No

---

## [Editor Report · Acceptance letter]

PONE-D-25-21587R2

PLOS One

Dear Dr. Yoo,

I'm pleased to inform you that your manuscript has been deemed suitable for publication in PLOS One. Congratulations! Your manuscript is now being handed over to our production team.

Kind regards,

on behalf of

Dr. Tianheng Shu

Academic Editor

PLOS One